# On the Learnability of Distribution Classes with Adaptive Adversaries

Tosca Lechner [1]   Alex Bie [* 2]   Gautam Kamath [* 1 3]

## Abstract

We consider the question of learnability of distribution classes in the presence of adaptive adversaries – that is, adversaries capable of intercepting the samples requested by a learner and applying manipulations with full knowledge of the samples before passing it on to the learner. This stands in contrast to oblivious adversaries, who can only modify the underlying distribution the samples come from but not their i.i.d. nature. We formulate a general notion of learnability with respect to adaptive adversaries, taking into account the budget of the adversary. We show that learnability with respect to additive adaptive adversaries is a strictly stronger condition than learnability with respect to additive oblivious adversaries.

## 1. Introduction

In the distribution learning problem, a learner receives i.i.d. samples from an unknown distribution $p$ belonging to a known class of distributions $\mathcal{C}$, and is tasked with producing an accurate estimate of $p$. Distribution learning is one of the most fundamental and well studied problems in learning theory (Kearns et al., 1994; Devroye & Lugosi, 2001); see also the survey of Diakonikolas (2016).

The above formulation of the problem is referred to as the *realizable* case – where the learner can take advantage of the strong prior knowledge that indeed, the unknown distribution $p$ they receive samples from is precisely a member of the distribution class $\mathcal{C}$. This assumption is dropped in the *agnostic* setting, where the learner must be able to handle receiving samples from a distribution outside of $\mathcal{C}$, but must produce an estimate close to the best approximation by a member of $\mathcal{C}$. An alternative formulation of this requirement is that the learner must be robust to an *oblivious adversary*:

> An *oblivious adversary* can modify the unknown distribution the learner's samples come from, with full knowledge of the learner's algorithm and $p$ itself, but cannot change the i.i.d. nature of the samples.

While all realizably learnable classes are agnostically learnable in the PAC setting (Vapnik & Chervonenkis, 1971; Haussler, 1992), the recent study of Ben-David et al. (2023) demonstrates that there is a separation in the distribution learning setting. They give an example of a realizably learnable class that is not learnable in the presence of an oblivous adversary. On the other hand, they also provide a positive result: a realizable learner for a class can be converted to a learner robust to oblivious adversaries restricted to only *additive* corruptions.[1]

It is a natural question to consider how the situation changes in the presence of an *adaptive adversary*:

> An *adaptive adversary* receives i.i.d. samples drawn from the unknown distribution $p$, and can modify individual samples with full knowledge of the samples, the learner's algorithm, and $p$ itself.

Indeed, the study of robustness with respect to adaptive adversaries is increasingly relevant to modern settings that examine machine learning algorithms from a worst-case, security perspective (Chen et al., 2017; Carlini & Wagner, 2017; Diakonikolas et al., 2019; Tramèr et al., 2020; Carlini et al., 2024).

Ben-David et al. (2023) focus entirely on the oblivious setting, and do not investigate the implication of their results to the adaptive setting. First, it is trivial to observe that their negative result (learnability does not imply robust learnability with an oblivious adversary) carries over to the adaptive case[2] This is because adaptive adversaries can simulate oblivious adversaries, and are thus stronger (see Table 1 for a full summary of the situation). The remaining unresolved question is whether their algorithmic results can be extended to the adaptive setting:

---

[1] Vector Institute [2] Google [3] University of Waterloo. Correspondence to: Tosca Lechner <tosca.lechner@vectorinstitute.ai>.

*Proceedings of the $42^{nd}$ International Conference on Machine Learning*, Vancouver, Canada. PMLR 267, 2025. Copyright 2025 by the author(s).

[1] This model is sometimes called *Huber contamination*.

[2] In the technical part of the paper we also show a slightly stronger version of this negative result for subtractive, which also holds for adversaries that only have access to $S$ but not to $p$. This result is shown via a separate proof technique and does not immediately follow from previous work.

|  | **Subtractive** | **Additive** |
|---|---|---|
| **Oblivious** | No (Ben-David et al., 2023, Theorem 2.1) | Yes (Ben-David et al., 2023, Theorem 1.7) |
| **Adaptive** | No $\Rightarrow$: implied by above | No *Answered by this work, Theorem 4.1* |

*Table 1.* Does learnability imply learnability with respect to [oblivious|adaptive], [subtractive|additive] adversaries?

*Are realizably learnable distributions learnable in the presence of an adaptive additive adversary?*

The present paper answers this question in the negative. We show that additive corruptions are strictly more powerful in the adaptive model than in the oblivious model.

To prove the result, we examine the relationship between subtractive and additive adversaries, and show that a general sufficient condition for the existence of adaptive subtractive adversaries also implies the existence of adaptive additive adversaries. This close relation between the additive and subtractive adversaries stands in contrast to the oblivious setting, where there subtractive adversaries are strictly more powerful than additive. We show that given an adaptive subtractive adversary, we can construct an adaptive additive adversary by inverting the subtractive adversary: instead of adding sample points that the subtractive learner would have deleted from a sample from a different distribution.

### 1.1. Results and techniques

We consider additive adversaries who can only add points, and subtractive adversaries can only remove points.[3]

Informally, a class is robustly learnable in the presence of an adversary if it admits a learner satisfying the following: given a sufficiently large (corrupted) sample, the learner is capable of driving error arbitrarily close to $\alpha \cdot \eta$, where $\eta$ is the fraction of samples added/removed by the adversary, and $\alpha$ is *any* absolute constant.

The following is our main result.

**Theorem 1.1** (Informal version of Theorem 4.1)**.** *There exists a class of distributions $\mathcal{C}$ that is realizably learnable, yet the class is not robustly learnable in the presence of an adaptive additive (or subtractive) adversary.*

In contrast, recall that the main algorithmic result of Ben-David et al. (2023) says that every realizably learnable class is also robustly learnable in the presence of an oblivious additive adversary.

---

[3]We defer a more formal definition of our adversary models to Section 3.2.

To obtain this result, we first develop a general technique for showing that a class is not learnable with respect to adaptive manipulations from an adversary $V$. This technique, based on a recent result of Ben-David & Lechner (2025), says the following:

**Theorem 1.2** (Informal version of Theorem 5.2)**.** *If for a class of distributions $\mathcal{C}$, there exists some $p \in \mathcal{C}$ and a meta-distribution $Q$ over elements of $\mathcal{C}$, such that*

1. *$d_{\mathrm{TV}}(p, q)$ is bounded below by some constant, for all $q$ in the support of $Q$; and*

2. *A sample $S \sim V(p^m)$ and a sample $S' \sim V(q^m)$ where $q \sim Q$ cannot be reliably distinguished,*

*then learning $\mathcal{C}$ with respect to adversary $V$ is hard.*

This result holds even if adversary $V$ does not have access to $p$ and can be found in Section 5. We use this result to show that in the adaptive case (in contrast to the oblivious case), additive and subtractive robustness are closely related (Section 6). In particular, we show that if the conditions (1.) and (2.) of Theorem 5.2 are satisfied by a class $\mathcal{C}$ and a subtractive adversary $V_{\mathrm{sub}}$, then $\mathcal{C}$ is not robustly learnable with respect to adaptive additive adversaries (Theorem 6.1).

Next, in Theorem 7.1 we show that fulfilling this condition for a subtractive adversary $V_{\mathrm{sub}}$ and a class $\mathcal{C}$ also implies the existence of a *single* adaptive additive adversary $V_{\mathrm{add}}$ that is successful against *every* learner for $\mathcal{C}$; we refer to such an adversary as a *universal adversary*.

The main result is first formally introduced in Section 4. We use a realizably learnable class $\mathcal{C}_g$ which was used to show a separation between agnostic and realizable learnability in (Ben-David et al., 2023). We then construct a subtractive adaptive adversary $V_{\mathrm{sub},\eta}$ and show that it meets the conditions (1.) and (2.) for Theorem 5.2. We then use our results from Section 6, to show that this also implies that $\mathcal{C}_g$ is not adaptively additively robustly learnable. In particular, we also show that there is both, a universal additive adversary $V_{\mathrm{add}}$ and a universal subtractive adversary $V_{\mathrm{sub},\eta}$ for $\mathcal{C}_g$. We introduce both the class $\mathcal{C}_g$ and an adaptive subtractive

adversary $V_{\text{sub},\eta}$ in Section 4, before delving into technical details.[4] We then motivate each of the subsequent sections, by the results we will get for $\mathcal{C}_g$ and the adversary $V_{\text{sub},\eta}$ within that section. We finally prove the main theorem at the end of the paper in Section 7.1.

## 2. Related work

As already mentioned, our work directly follows up on, and addresses an open problem of, Ben-David et al. (2023). Their work shows that learnability implies robust learnability under an oblivious additive adversary but not under an oblivious subtractive adversary. They explicitly asked whether their algorithms which are effective under an oblivious additive adversary can be extended to handle an adaptive additive adversary. We answer this question in the negative: learnability does not imply robust learnability under an adaptive additive adversary.

Recent works of Blanc, Lange, Malik, Tan, and Valiant (Blanc et al., 2022; Blanc & Valiant, 2024) also study the relationship of adaptive and oblivious adversaries. They show impressively generic results: for a broad range of statistical tasks, given an algorithm that works against an oblivious adversary, this can be converted to an algorithm that works against an adaptive adversary by simply drawing a larger dataset and randomly subsampling from it. This seems to suggest that any distribution which is learnable under an oblivious adversary should also be learnable under an adaptive adversary, which would contradict our main result. However, there is no contradiction: the size of the "larger dataset" their approach requires depends logarithmically on the support size of the distribution, and we focus on distributions with unbounded support. Thus our results imply that some dependence on the domain size is unavoidable for this reduction to go through for the task of distribution learning.

One recent work of Canonne et al. (2023) shows a gap between the sample complexity of robust Gaussian mean testing with adaptive and oblivious adversaries: they show that the adaptive adversary is strictly more powerful, necessitating an increase in the sample complexity. Their focus is on a testing problem, whereas we study a distribution learning problem. They show a polynomial gap in the sample complexity for a natural problem, whereas we show an infinite gap in the sample complexity of a somewhat contrived problem.

Robustness is a traditional topic of study within the field of Statistics, see, for example, the classic works (Tukey, 1960; Huber, 1964). Within Computer Science, distribution learning has been studied since the work of Kearns et al. (1994), inspired by Valiant's PAC learning model (Valiant, 1984).

---

[4]The adversary $V_{\text{sub},\eta}$ does not require knowledge of the ground-truth distribution $p$.

Many subsequent works have studied algorithms for learning particular classes of distributions, see, e.g., (Chan et al., 2013; 2014a;b; Li & Schmidt, 2017; Ashtiani et al., 2020). A recent line of work, initiated by (Diakonikolas et al., 2016; Lai et al., 2016), studies computationally-efficient algorithms for robust estimation of particular classes of multivariate distributions, see, e.g., (Diakonikolas et al., 2017; Steinhardt et al., 2018; Diakonikolas et al., 2018; Kothari et al., 2018; Hopkins & Li, 2018; Diakonikolas et al., 2019; Liu & Moitra, 2021; 2022; Bakshi et al., 2022; Jia et al., 2023) and (Diakonikolas & Kane, 2022) for a reference. We focus on understanding broad and generic connections between learnability and robust/agnostic learnability, without consideration for computation, in contrast to those works that focus on computation and particular classes of distributions.

## 3. Setup

### 3.1. Preliminaries

We consider learning over a *domain* $\mathcal{X}$. We denote the set of all *distributons over* $\mathcal{X}$ as $\Delta(\mathcal{X})$. We assume an i.i.d. generating process of sample sets $S$. If $S$ is a sample of size $m$ i.i.d. drawn from a distribution $p$, we will denote this as $S \sim p^m$. Furthermore, we note that we consider samples $S$ to be multi-sets, that is, we consider samples to be randomly shuffled/order invariant, but assume that repeated elements are counted. For example, we assume that $S = \{0,1,1\} = \{1,0,1\}$, but $\{1,0,0\} \neq \{1,0\}$. In a slight abuse of notation we will use set-operations on samples $S$, again assuming that elements are repeated. That is $\{a,b,b,c\} \cup \{a,b,d,d\} = \{a,a,b,b,c,d,d\}$ and $\{a,b,b,c\} \setminus \{a,b,d,d\} = \{b,c\}$.

We let $\mathcal{X}^* = \bigcup_{i=0}^{\infty} \mathcal{X}^i$, where $\mathcal{X}^m$ is the set of multi-sets over $\mathcal{X}$ of size $m$. We usually refer to learning with respect to a concept class of distributions $\mathcal{C} \subset \Delta(\mathcal{X})$. Furthermore, we consider distribution learning with respect to total variation distance $d_{\text{TV}} : \Delta(\mathcal{X}) \times \Delta(\mathcal{X}) \to [0,1]$ defined by

$$d_{\text{TV}}(p,q) = \sup_{B \subset \mathcal{X} : B \text{ measurable}} |p(B) - q(A)| =$$
$$= \frac{1}{2} \int_{x \in \mathcal{X}} |dp(x) - dq(x)|.$$

We study the PAC learnability of distribution classes in the presence of adaptive adversaries. We start by giving the definition of PAC learnability of a class of distribution in the realizable case, i.e., without the presence of any adversary.

**Definition 3.1** ((Realizable) PAC learnability). A class of distributions $\mathcal{C}$ is (realizably) PAC learnable, if there exists a learner $A$ and a sample complexity function $m_{\mathcal{C}}^{\text{re}} : (0,1)^2 \to \mathbb{N}$, such that for every $p \in \mathcal{C}$ and every $\varepsilon, \delta \in (0,1)$ and

every $m \geq m_{\mathcal{C}}^{\mathrm{re}}(\varepsilon, \delta)$, with probability $1 - \delta$,

$$d_{\mathrm{TV}}(A(S), p) \leq \varepsilon$$

where $S \sim p^m$.

## 3.2. Adaptive adversaries

An *adaptive adversary* is a function $V : \mathcal{X}^* \to \mathcal{X}^*$ from samples to samples.[5] We allow this function to be randomized. We refer to the probability measure of $V(S)$ by $p_{V(S)}$. When considering $S$ to be generated by some distribution $p^m$, we will sometimes refer to the distribution of $V(S)$ by $V(p^m)$ in a slight abuse of notation.

We now introduce two main classes of adaptive adversaries, *additive adversaries*, who can only add additional sample points $S_1'$ to the sample, i.e., $V_{\mathrm{add}}(S) = S \cup S_1'$ and *subtractive adversaries*, who can only delete some sample points $S_2'$ from the input sample, i.e., $V_{\mathrm{sub}}(S) = S \setminus S_2'$. We will also introduce a notion of budget, which limits the amount of manipulation of an adversary.

**Additive adaptive adversaries** We say that adaptive adversary $V$ is *additive*, if for every $S \in \mathcal{X}^*$, we have $S \subset V(S)$. We denote the class of all adaptive additive adversaries with $\mathcal{V}_{\mathrm{add}}$.

**Subtractive adaptive adversaries** We say that adaptive adversary $V$ is *subtractive*, if for every $S \in \mathcal{X}^*$, we have $V(S) \subset S$. We denote the class of all subtractive adaptive adversaries with $\mathcal{V}_{\mathrm{sub}}$.

In the presence of an adversary, a learner $A$ does not have direct access to an i.i.d. generated sample $S \sim p^m$ from ground-truth distribution $p \in \mathcal{C}$, but only indirect access via a manipulated sample $V(S)$.

In general, we can not hope to approximate the ground-truth distribution $p$ to a total variation distance up to any $\varepsilon > 0$, but rather, we have to figure in the budget of the adversary. The budget of an adversary is some function $\mathrm{budget} : \mathcal{X}^{*\mathcal{X}^*} \times \mathbb{N} \to [0, 1]$ and models their manipulation capabilities.

In cases, where the power of the adversary amounts to either adding or deleting instances, but not changing instances in any other way, the budget amounts to the edit distance. In general, the budget for an adaptive additive adversary is thus defined by:

---

[5] Adaptive adversaries may also make use knowledge of the underlying sample generating distribution $p$. We omit this option for simplicity. Indeed, our main result is slightly stronger than stated: we show that adversaries that do not make use of knowledge of $p$ suffice to prevent learning.

$\mathrm{budget}^{\mathrm{add}} : \mathcal{X}^{*\mathcal{X}^*} \times \mathbb{N} \to [0, 1]$ is defined by:

$$\mathrm{budget}^{\mathrm{add}}(V, m) = \sup_{S \in \mathcal{X}^m} \frac{|V(S)| - |S|}{m}.$$

Similarly, the budget for a subtractive adversary is defined by

$$\mathrm{budget}^{\mathrm{sub}}(V, m) = \sup_{S \in \mathcal{X}^m} \frac{|S| - |V(S)|}{m}.$$

In this work, we only consider adversaries that have fixed budgets and furthermore that those budgets are constant. In particular, we will assume, that for both subtractive and additive adversaries $V$, for all $m \in \mathbb{N}$ and all $S_1, S_2 \in \mathcal{X}^m$ we have $|V(S_1)| = |V(S_2)|$. Furthermore, we assume that for every adversary $V$ there is a constant $\mathrm{budget}(V) = \eta$, such that for every $m \in \mathbb{N}$:

$$\eta m - 1 < m \cdot \mathrm{budget}(V, m) \leq \eta m.$$

We can now define robust PAC learning with respect to a specific adaptive adversary.

**Definition 3.2** (adaptive $\alpha$-robust with respect to adversary $V$). Let $\alpha \geq 1$. A class of distributions $\mathcal{C}$ is adaptively $\alpha$-robustly learnable w.r.t. adversary $V$, if there exists a learner $A$ and a sample complexity function $m_{\mathcal{C}}^{V, \alpha} : (0, 1)^2 \to \mathbb{N}$, such that for every $p \in \mathcal{C}$, every $\varepsilon, \delta \in (0, 1)$ and every sample size $m \geq m_{\mathcal{C}}^{V, \alpha}(\varepsilon, \delta)$ with probability $1 - \delta$,

$$d_{\mathrm{TV}}(A(V(S)), p) \leq \alpha \cdot \mathrm{budget}(V) + \varepsilon.$$

If a class $\mathcal{C}$ is not $\alpha$-robustly learnable with respect to adversary $V$, we say that $V$ is a *universal $\alpha$-adversary* for $\mathcal{C}$, as every learner for $\mathcal{C}$ fails against $V$.

In general, learners want to defend against more than one potential adversary, since a priori, they may not know the adversary's strategy. Thus in the next definition we define learnability with respect to a class of adversaries. One can also think of this as a strengthening of the adversary, as here the learner needs to choose the learning rule first, and the adversary can adapt their strategy to the selected learning rule.

**Definition 3.3.** Let $\alpha \geq 1$. A class of distributions $\mathcal{C}$ is adaptively $\alpha$-robustly learnable w.r.t. a class of adversaries $\mathcal{V}$, if there exists a learner $A$ and a sample complexity function $m_{\mathcal{C}}^{\mathcal{V}, \alpha} : (0, 1)^2 \to \mathbb{N}$, such that for every $p \in \mathcal{C}$ and every $V \in \mathcal{V}$ and every $\varepsilon, \delta \in (0, 1)$ and every $m \geq m_{\mathcal{C}}^{\mathcal{V}, \alpha}(\varepsilon, \delta)$, with probability $1 - \delta$,

$$d_{\mathrm{TV}}(A(V(S)), p) \leq \alpha \cdot \mathrm{budget}(V) + \varepsilon.$$

We say a class $\mathcal{C}$ is *adaptively additively $\alpha$-robustly learnable* if $\mathcal{C}$ is $\alpha$-robustly learnable with respect to the class of adversaries $\mathcal{V}_{\mathrm{add}}$. We say a class $\mathcal{C}$ is *adaptively subtractively $\alpha$-robustly learnable* if $\mathcal{C}$ is $\alpha$-robustly learnable with respect to the class of adversaries $\mathcal{V}_{\mathrm{sub}}$.

# 4. A realizable class with an adaptive adversary

In this section, we formally introduce the main result of this paper: there are classes of distributions which are learnable in the realizable case but not learnable in the presence of either adaptive additive or adaptive subtractive adversaries. Since learnability in the realizable setting also implies learnability with an oblivious additive adversary (Ben-David et al., 2023), this result also implies a separation between learnability with respect to oblivious and adaptive adversaries in the additive case.

In this section, we give the formal statement of the main result (Theorem 4.1). We also describe Theorem 4.1's subject distribution class $\mathcal{C}_g$, and argue it is realizably learnable. Finally, we introduce a subtractive adversary $V_{\mathrm{sub},\eta}$ for the class. In subsequent sections we will use this adversary to prove the remaining parts of Theorem 4.1, which we now state.

**Theorem 4.1.** *For every superlinear function $g : \mathbb{R} \to \mathbb{R}$, there is class $\mathcal{C}_g$, such that*

- *$\mathcal{C}_g$ is realizably learnable with sample complexity $m_{\mathcal{C}_g}^{\mathrm{re}}(\varepsilon, \delta) \leq \log\left(\frac{1}{\delta}\right) g\left(\frac{1}{\varepsilon}\right)$*

- *For every $\alpha \geq 1$, $\mathcal{C}_g$ is not adaptively additive $\alpha$-robustly learnable. Moreover, for every $\alpha \geq 1$, there is an adaptive additive adversary $V_{\mathrm{add}}$, that is a universal $\alpha$-adversary for $\mathcal{C}_g$.*

- *For every $\alpha \geq 1$, $\mathcal{C}_g$ is not adaptively subtractively $\alpha$-robustly learnable. Moreover, for every $\alpha \geq 1$, there is an adaptive subtractive adversary $V_{\mathrm{sub}}$, that is a universal $\alpha$-adversary for $\mathcal{C}_g$.*

**Introducing class $\mathcal{C}_g$.** Theorem 4.1 uses the class $\mathcal{C}_g$ from Ben-David et al. (2023) that was used to show a separation between realizable and agnostic learning.[6]

For this let $\{B_i \subset \mathbb{N} : B_i \text{ is finite}\}$ be an enumeration of all finite subsets indexed by $i \in \mathbb{N}$. Now for constants $j, k \in \mathbb{N}$, we define the following distribution as a mixture of two point masses $\delta_{(0,0)}$ and $\delta_{(i,j)}$ centered at $(0,0)$ and $(i,j)$ respectively, and a uniform distribution over the set $B_i$ denoted by $U_{B_i}$:

$$p_{i,j,k} = \left(1 - \frac{1}{j}\right) \delta_{(0,0)} + \left(\frac{1}{j} - \frac{1}{k}\right) U_{B_i \times \{2j+1\}} + \frac{1}{k}\delta_{(i,2j+2)}.$$

Now for a function $g : \mathbb{N} \to \mathbb{N}$, we define the class

$$\mathcal{C}_g = \{p_{i,j,g(j)} : i \in \mathbb{N}, j \in \mathbb{N}\}.$$

---

[6]This class is denoted $\mathcal{Q}_g$ in (Ben-David et al., 2023).

**$\mathcal{C}_g$ is realizably learnable.** We first note that this class is realizably learnable, using results from Ben-David et al. (2023).

**Lemma 4.2** (Claim 3.2 from Ben-David et al. (2023)). *For every monotone function $g : \mathbb{N} \to \mathbb{N}$, the class $\mathcal{C}_g$ is learnable with sample complexity*

$$m_{\mathcal{C}_g}^{\mathrm{re}}(\varepsilon, \delta) \leq \log\left(\frac{1}{\delta}\right) g\left(\frac{1}{\varepsilon}\right).$$

The realizable learner is based on the idea, that for every distribution the learner only needs to observe a unique indicator bit in order to perfectly identify the distribution.

**Subtractive adversary $V_{\mathrm{sub},\eta}$ for $\mathcal{C}_g$** We will now introduce a subtractive adversary $V_{\mathrm{sub},\eta}$ for $\mathcal{C}_g$. The properties of this adversary will then be used in later sections of the paper to show that $\mathcal{C}_g$ is neither adaptive additive nor adaptive subtractive robustly learnable.

For a sample $S$, we partition $S$ into the non-indicators $\mathrm{constants}(S) = \{(0,0) \in S\}$ and $\mathrm{odds}(S) = \{(o, 2j + 1) \in S : o, j \in \mathbb{N}\}$ and indicators $\mathrm{ind}(S) = \{(i, 2j + 2) \in S : i, j \in \mathbb{N}\}$, i.e. $S = \mathrm{constants}(S) \uplus \mathrm{odds}(S) \uplus \mathrm{ind}(S)$. We further denote non-indicators $\mathrm{noind}(S) = \mathrm{constants}(S) \cup \mathrm{odds}(S)$. Note that $S$, $\mathrm{constants}(S)$, $\mathrm{odds}(S)$, $\mathrm{noind}(S)$, and $\mathrm{ind}(S)$ are all multisets and thus repetitions of elements are counted.

We now define the subtractive adversary $V_{\mathrm{sub},\eta} : \mathcal{X}^* \to \mathcal{X}^*$ as the adversary that removes from $S$ as many elements belonging to $\mathrm{ind}(S)$ as possible while meeting the budget constraint $\mathrm{budget}(V_{\mathrm{sub},\eta}) \leq \eta$. If there are no elements in $\mathrm{ind}(S)$ left to remove, $V_{\mathrm{sub},\eta}$ chooses to remove elements randomly to match the budget. Formally,

$$V_{\mathrm{sub},\eta}(S) = \begin{cases} \mathrm{choose}(\mathrm{noind}(S), (1 - \eta)|S|) \\ \qquad, \text{if } |\mathrm{ind}(S)| \leq \eta|S| \\ \mathrm{noind}(S) \,\cup \\ \mathrm{choose}(\mathrm{ind}(S), |\mathrm{ind}(S)| - \eta|S|) \\ \qquad, \text{if } |\mathrm{ind}(S)| > \eta|S| \end{cases},$$

where $\mathrm{choose}(S, n)$ is the random variable, which selects a uniformly chosen random subset $S' \subset S$ of size $n$. It is easy to see that $V_{\mathrm{sub},\eta}$ is a subtractive adversary with $\mathrm{budget}^{\mathrm{sub}}(V_{\mathrm{sub},\eta}) = \eta$. We note, that the definition of this adversary $V_{\mathrm{sub},\eta}$ does not require any knowledge of the ground-truth distribution $p$. While this adversary is in idea similar to the subtractive oblivious adversaries that were shown to be successful against learners in (Ben-David et al., 2023), these past results do not yet show that $V_{\mathrm{sub},\eta}$ is a successful adversary for $\mathcal{C}_g$. In the following section, we introduce the notion of an adversary *successfully confusing*

samples generated from members of $\mathcal{C}_g$ to show that an adversary is a universal $\alpha$-adversary. We then show that for every $\alpha \geq 1$, there exists $\eta \in (0, 1)$, such that $V_{\mathrm{sub}, \eta}$ satisfies this notion, and hence is indeed a universal $\alpha$-adversary for $\mathcal{C}_g$. We then also show that this condition implies that $\mathcal{C}_g$ is not adaptive additive $\alpha$-robustly learnable (Section 6). Moreover, using the same condition, we show that for every $\alpha \geq 1$, there exists a universal additive $\alpha$-adversary for $\mathcal{C}_g$ (Section 7). The proof of Theorem 4.1 can be found in Section 7 at the end of the paper.

## 5. General technique for showing distribution classes cannot be learned adaptively

In this section, we will show a general lower bound for learning in the presence of adaptive adversaries. We introduce the notion of an adversary $V$ or a pair of adversaries $(V_1, V_1)$ *successfully confusing* samples generated from a class $\mathcal{C}$ and show that this condition is sufficient to show a class $\mathcal{C}$ cannot be learned in the presence of adversary $V$ (or pair of adversaries $(V_1, V_2)$, respectively). Essentially, the result shows that if adversaries can make samples from certain random distributions defined over $\mathcal{C}$ sufficiently indistinguishable, the adversary can also fool any learner. To state the definition of successfuly confusing a $\mathcal{C}$-generated sample, we introduce some notation.

For a distribution $Q$ over a class of distributions $\mathcal{C}$, let $|Q|^m$ denote the distribution over $\mathcal{X}^m$ that results from first sampling $q \sim Q$ and then $S \sim q^m$. Furthermore, let $\mathrm{supp}(Q)$ denote the support of $Q$. In the following we will pick distributions $p \in \mathcal{C}$ and $\mathcal{Q} \in \Delta(\mathcal{C})$ such that for every $q \in \mathrm{supp}(Q)$ the total variation distances $d_{\mathrm{TV}}(p, q)$ are upper bounded by some constant.

If for such distributions $p$ and $Q$ there are adaptive adversaries $V_1, V_2$ that can make the sample distributions $V_1(p^m)$ and $V_2(|Q|^m)$ sufficiently hard to distringuish, then the class $\mathcal{C}$ is not robustly learnable with respect to $\{V_1, V_2\}$. To show this, we introduce the following notion:

**Definition 5.1.** Let $\mathcal{C}$ be a class of distributions. We say a pair of adversaries $(V_1, V_2)$ *successfully $(\gamma, \zeta)$-confuses $\mathcal{C}$-generated samples of size $m$* if there is a distribution $p \in \mathcal{C}$ and a meta-distribution $Q \in \Delta(\mathcal{C})$ with

- for all $q \in \mathrm{supp}(Q)$ we have $d_{\mathrm{TV}}(p, q) > \gamma$

- $d_{\mathrm{TV}}\left(V_1(|Q|^m), V_2(p^m)\right) < \zeta$.

If $V_1 = V_2$, we also say $V_1$ successfully $(\gamma, \zeta)$-confuses $\mathcal{C}$-generated samples of size $m$.

Successful adversaries have large $\gamma$ and small $\zeta$. We now state the main theorem of this section which shows that if adversaries successfully confuse $\mathcal{C}$-generated samples for

every size $m$, then $\mathcal{C}$ is not robustly learnable with respect to those adversaries.

**Theorem 5.2.** *Let $\mathcal{C}$ be a class of distributions and $\mathcal{V} \supset \{V_1, V_2\}$ a set of adaptive adversaries with budgets $\mathrm{budget}(V_1) = \eta_1$ and $\mathrm{budget}(V_2) = \eta_2$. Let $\gamma', \zeta \in (0, 1)$ and define*

$$\gamma = 2\alpha \max\{\eta_1, \eta_2\} + 2\gamma'.$$

*If for every $m \in \mathbb{N}$ the pair of adversaries $(V_1, V_2)$ successfully $(\gamma, \zeta)$-confuses $\mathcal{C}$-generated samples of size $m$, then $\mathcal{C}$ is not $\alpha$-robustly learnable with respect to $\mathcal{V}$.*

*Furthermore, if $V_1 = V_2$, then $V_1$ is a universal $\alpha$-adversary for $\mathcal{C}$.*

We show this result by the following lemma which makes the same claim for a fixed sample size $m$.

**Lemma 5.3.** *Let $\mathcal{C}$ be a class of distributions, let $A$ be a learner and $(V_1, V_2)$ a pair of adversaries that successfully $(\gamma, \zeta)$-confuses $\mathcal{C}$-generated samples of size $m$.*

*Then for every learner $A$, there is $r \in \mathcal{C}$, such that*

$$\mathcal{P}_{S \sim r^m}\left[d_{\mathrm{TV}}(A(V_1(S)), r) > \frac{\gamma}{2}\right] \geq \frac{1}{2} - \frac{\zeta}{2}$$

*or*

$$\mathcal{P}_{S \sim r^m}\left[d_{\mathrm{TV}}(A(V_2(S)), r) > \frac{\gamma}{2}\right] \geq \frac{1}{2} - \frac{\zeta}{2}.$$

Lemma 5.3 is a corollary of a result of Ben-David & Lechner (2025), which makes the connection between the indistinguishability of the sample distributions $|Q|^m$ and $p^m$, and hardness of learning. For completeness, we include the full proof of Lemma 5.3 in Appendix A.1, but we note that it follows the exact same argument as the proof of the cited result.

**Lemma 5.4** (Lemma 4 of Ben-David & Lechner (2025)). *Let $\mathcal{C}_1, \mathcal{C}_2$ be such that for all $q \in \mathcal{C}_1$ and all $p \in \mathcal{C}_2$, we have $d_{\mathrm{TV}}(p, q) > \gamma$. If there is a distribution $Q$ over $\mathcal{C}_1$ and $p \in \mathcal{C}_2$ such that for $\zeta \in (0, 1/2)$ we have $d_{\mathrm{TV}}(|Q|^m, p^m) < \zeta$, then for ever learner $A$, there is $r \in \mathcal{C}_1 \cup \mathcal{C}_2$, such that*

$$\mathcal{P}_{S \sim r^m}\left[d_{\mathrm{TV}}(A(S), r) > \frac{\gamma}{2}\right] \geq \frac{1}{2} - \frac{\zeta}{2}.$$

The proof of both Lemma 5.3 and Theorem 5.2 can be found in the appendix. Furthermore, in Appendix B we give an example which illustrates how these lemmas can be applied. A similar intuition to that example is used in the next section to show that $V_{\mathrm{sub}, \eta}$ successfully confuses $\mathcal{C}_g$.

### 5.1. $V_{\mathrm{sub}, \eta}$ is universal adversary for $\mathcal{C}_g$

In this subsection we show that for every $\alpha \geq 1$ there is $\eta \in (0, 1)$, such that $V_{\mathrm{sub}, \eta}$ is a universal $\alpha$-adversary for

$\mathcal{C}_g$ (recall the constructions of these objects as described in Section 4). We will first show that $V_{\mathrm{sub},\eta}$ successfully confuses $\mathcal{C}_g$-generated samples and then apply Theorem 5.2.

**Lemma 5.5.** *For every $\alpha \geq 1$, there is $\eta, \gamma' \in (0,1)$, such that for the subtractive adversary $V_{\mathrm{sub},\eta}$ the following holds: For every $m \geq 1$ there are distributions $p \in \mathcal{C}_g$ and $Q \in \Delta(\mathcal{C}_g)$ such that*

- *for every $q \in \mathrm{supp}(Q)$:*

$$d_{\mathrm{TV}}(p,q) \geq 4\alpha \cdot \mathrm{budget}(V_{\mathrm{sub},\eta}) + 4\gamma'$$

- $d_{\mathrm{TV}}(V_{\mathrm{sub},\eta}(p^m), V_{\mathrm{sub},\eta}(|Q|^m)) \leq \frac{1}{8}$.

*Proof sketch.* We first note that for $p = p_{i,j,k}$ where $|B_i| = 2^{2^n}$ is arbitrary and $D_{i,n,j,k} = \{p_{i',j,k} : B_{i'} \subset B_i : |B_{i'}| = 2^n\}$, we have that for every $q \in D_{i,n,j,k}$

$$d_{\mathrm{TV}}(p,q) \geq \left(\frac{1}{j} - \frac{1}{k}\right) d_{\mathrm{TV}}(U_{B_i \times 2j}, U_{B_{i'} \times 2j})$$
$$\geq \frac{1}{2}\left(\frac{1}{j} - \frac{1}{k}\right).$$

Furthermore, consider $Q = U_{D_{i,n,j,k}}$. We note that the distributions $V_{\mathrm{sub},\eta}(p^m)$ and $V_{\mathrm{sub},\eta}(|Q|^m)$ are both distributions over multisets $S' = \mathrm{constants}(S') \cup \mathrm{odds}(S') \cup \mathrm{ind}(S')$. We further note that the distributions of the count of each of those subsets are the same for both $S' \sim V_{\mathrm{sub},\eta}(p^m)$ and $S' \sim V_{\mathrm{sub},\eta}(|Q|^m)$. Furthermore, since all elements of $\mathrm{constants}(S')$ are the same, we also have that the probability distributions of $\mathrm{constants}(S')$ are the same for both $S' \sim V_{\mathrm{sub},\eta}(p^m)$ and $S' \sim V_{\mathrm{sub},\eta}(|Q|^m)$. We also observe that the distributions for $\mathrm{odds}(S')$ conditioned on $S' \sim V_{\mathrm{sub},\eta}(p^m)$ and $S' \sim V_{\mathrm{sub},\eta}(|Q|^m)$ are the same, if $\mathrm{odds}(S')$ does not contain any repeated elements. Lastly, we note that while the indicators of samples from $p$ and samples from $Q$ differ, the adversary $V_{\mathrm{sub},\eta}$ deletes all these elements, if $|\mathrm{ind}(S)|$ does not exceed $\eta|S|$.

Taking all of these observations together, we can bound the total variation distance in terms of repeating elements in $\mathrm{odds}(S)$ and the probability that $|\mathrm{ind}(S)|$ exceeds the budget of the adversary.

$$d_{\mathrm{TV}}(V_{\mathrm{sub},\eta}(p^m), V_{\mathrm{sub},\eta}(|Q|^m))$$
$$\leq \mathbb{P}_{S \sim p^m}[\mathrm{odds}(S) \text{ contains repeated elements}]$$
$$+ \mathbb{P}_{S \sim |Q|^m}[\mathrm{odds}(S) \text{ contains repeated elements}]$$
$$+ \mathbb{P}_{S \sim p^m}[|\mathrm{ind}(S)| > \eta|S|]$$
$$+ \mathbb{P}_{S \sim |Q|^m}[|\mathrm{ind}(S)| > \eta|S|].$$

The first two terms can each be upper bounded by $1 - \left(1 - \frac{m}{2^n}\right)^m$ using the birthday paradox. The last two terms

can each be upper bounded by

$$\mathbb{P}_{S \sim p^m}[|\mathrm{ind}(S)| > \eta|S|] + \mathbb{P}_{S \sim |Q|^m}[|\mathrm{ind}(S)| > \eta|S|]$$
$$\leq \frac{\mathbb{E}_{S \sim p^m}[|\mathrm{ind}(S)|]}{\eta|S|} + \frac{\mathbb{E}_{S \sim |Q|^m}[|\mathrm{ind}(S)|]}{\eta|S|} \leq 2 \cdot \frac{\frac{m}{k}}{\eta m} = \frac{2}{k\eta},$$

using Markov's inequality.

Since we are considering distributions in $\mathcal{C}_g$, we note that the distributions $D_{i,n,j,k}$ need to be of the form $D_{i,n,j,g(j)}$. We now want to argue that for appropriate choices of $j$ and $\eta$ (both independent of $m$), as well as for $n$ given $m$, the inequalities of the theorem are satisfied. First, we note that since $g$ grows faster than linear, it is possible to pick $j$ in such a way that it satisfies the inequality

$$g(j) \geq 1024c\alpha j.$$

Given such $j$, we then pick $\eta = \frac{32}{g(j)}$ and $\gamma' = \frac{\alpha}{g(j)}$. This ensures, that for every $n \in \mathbb{N}$ and every $q \in D_{i,n,j,g(j)}$, we get

$$d_{\mathrm{TV}}(p,q) \geq \frac{1}{2}\left(\frac{1}{j} - \frac{1}{g(j)}\right)$$
$$= 4\alpha\eta + 4\gamma'.$$

Then, for every $m \geq 1$, if we choose $n \geq \frac{m}{1 - (1 - \frac{1}{32})^{1/m}}$, we get

$$d_{\mathrm{TV}}(V_{\mathrm{sub},\eta}(p^m), V_{\mathrm{sub},\eta}(|Q|^m))$$
$$\leq \frac{2}{g(j) \cdot \frac{32}{g(j)}} + 2\left(1 - \left(1 - \frac{m}{2^n}\right)^m\right)$$
$$\leq \frac{1}{8}.$$

$\square$

The full proof with all the calculations can be found in the appendix.

## 6. Subtractive versus additive adaptive adversaries

In this section, we will show that unlike in the oblivious case, additive and subtractive adaptive adversaries are closely related. In particular, we show that if there is a universal subtractive adversary $V_{\mathrm{sub}}$ that successfully $(\gamma, \zeta)$-confuses $\mathcal{C}$-generated samples of size $m$, then there is a pair of additive adversaries $(V_{\mathrm{add},p^m}, V_{\mathrm{add},|Q|^m})$ that also successfully $(\gamma, \zeta)$-confuses $\mathcal{C}$-generated samples of size $m$.

**Theorem 6.1.** *Let $\mathcal{C}$ be a class of distributions. Let $V_{\mathrm{sub}}$ be an adaptive subtractive adversary. Let $\zeta \in (0,1)$ be a constant, $p \in \mathcal{C}$ a distribution, and $Q$ a distribution over elements in $\mathcal{C}$ such that $d_{\mathrm{TV}}(V_{\mathrm{sub}}(|Q|^m), V_{\mathrm{sub}}(p^m)) < \zeta$.*

*Then there are additive adversaries $V_{\mathrm{add},p^m}$ and $V_{\mathrm{add},|Q|^m}$ with $d_{\mathrm{TV}}(V_{\mathrm{add},p^m}(|Q|^m), V_{\mathrm{add},|Q|^m}(p^m)) < \zeta$. Furthermore, if $V_{\mathrm{sub}}$ has a fixed constant budget of $\eta < \frac{1}{2}$, then both $V_{\mathrm{add},|Q|^m}$ and $V_{\mathrm{add},p^m}$ have fixed constant budgets of no more than $\eta$.*

In other words, we argue that, if there is an element $p \in \mathcal{C}$ and a meta distribution $Q$ over $\mathcal{C}$ that can be made hard to distinguish by a subtractive adversary $V_{\mathrm{sub}}$, i.e., $d_{\mathrm{TV}}(V_{\mathrm{sub}}(|Q|^m), V_{\mathrm{sub}}(p^m)) < \zeta$, then this adversary can be used to construct additive adversaries $V_{\mathrm{add},|Q|^m}$ and $V_{\mathrm{add},p^m}$, such that the resulting additive sample distributions are equally close, i.e., $d_{\mathrm{TV}}((V_{\mathrm{add},|Q|^m}(p^m), V_{\mathrm{add},p^m}(|Q|^m)) < \zeta$.

*Proof sketch.* While adversaries act on samples drawn from two different distributions to make the manipulated sample distributions close, we will first give an illustration in terms of manipulating simple point-sets $S_{p^m} \sim p^m$ and $S_{|Q|^m} \sim |Q|^m$.

Roughly speaking, a successful subtractive adversary can remove part of the first sample $S_{p^m}$ and part of the second sample $S_{|Q|^m}$ to leave behind a common set $S_{\mathrm{core}} = V_{\mathrm{sub}}(S_{p^m}) = V_{\mathrm{sub}}(S_{|Q|^m}) \subset S_{p^m} \cap S_{|Q|^m}$. We can view the generative process of $S_{p^m}$ to be a sample $S_{\mathrm{core}}$ combined with a sample $S_{p^m} \setminus S_{\mathrm{core}}$, and the generative process of $S_{|Q|^m}$ to be a sample $S_{\mathrm{core}}$ combined with a sample from $S_{|Q|^m} \setminus S_{\mathrm{core}}$. Hence to confuse the learner, the additive adversaries just needs to add the "opposite piece," i.e., $V_{\mathrm{add},Q^m}$ is mapping $S_{p^m} = S_{\mathrm{core}} \cup (S_{p^m} \setminus S_{\mathrm{core}})$ to $S_{\mathrm{core}} \cup (S_{p^m} \setminus S_{\mathrm{core}}) \cup (S_{|Q|^m} \setminus S_{\mathrm{core}})$ and $V_{\mathrm{add},p^m}$ is mapping $S_{Q^m} = S_{\mathrm{core}} \cup (S_{Q^m} \setminus S_{\mathrm{core}})$ to $S_{\mathrm{core}} \cup (S_{Q^m} \setminus S_{\mathrm{core}}) \cup (S_{p^m} \setminus S_{\mathrm{core}})$. Thus, if a pair of samples $S_{p^m}$ and $S_{|Q|^m}$ could be made indistinguishable by a subtractive adversary $V_{\mathrm{sub}}$, then they can also be made indistinguishable by a pair of adversaries $V_{\mathrm{add},|Q|^m}$ and $V_{\mathrm{add},p^m}$.

Now we want to lift this intuition from samples to a more rigorous discussion of distributions. We note that if the subtractive adversary $V_{\mathrm{sub}}$ is successfully confusing distributions $p^m$ and $|Q|^m$, then there is a distribution $p_{\mathrm{core}}$ of common sets $S_{\mathrm{core}} \sim p_{\mathrm{core}}$ that is close to both $V_{\mathrm{sub}}(p^m)$ and $V_{\mathrm{sub}}(|Q|^m)$ in total variation distance. Now, due to $V_{\mathrm{sub}}(p^m)$ and $p_{\mathrm{core}}$ being close, most samples in $S_{p^m} \sim p^m$ can successfully be generated in an alternative way by first sampling $S_{\mathrm{core}} \sim p_{\mathrm{core}}$ and then reversing the subtractive adversary $V_{\mathrm{sub}}$ to generate $S_{p^m} = S_{\mathrm{core}} \cup (S_p \setminus S_{\mathrm{core}}) = V_{\mathrm{sub},p^m}^{-1}(S_{\mathrm{core}})$. In order to successfully reverse $V_{\mathrm{sub}}$ we also need access to a prior distribution that generated the input samples for the adversary $V_{\mathrm{sub}}$. If we have a prior distribution $p^m$, we can define $V_{\mathrm{sub},p^m}^{-1}(S_{\mathrm{core}})$ as the conditional distribution of $S$, given $V_{\mathrm{sub}}(S) = S_{\mathrm{core}}$, i.e., for

every measurable subset $B \subset \mathcal{X}$:

$$\mathbb{P}_{S \sim V_{\mathrm{sub},p^m}^{-1}(S_{\mathrm{core}})}[S \in B] = $$
$$\mathbb{P}_{S \sim p^m}[S \in B | V_{\mathrm{sub}}(S) = S_{\mathrm{core}}].$$

Similarly, we can make the same observations for $|Q|^m$ and define the reversed adversary $V_{\mathrm{sub},|Q|^m}^{-1}$ equivalently. The additive adversaries $V_{\mathrm{add},|Q|^m}$ and $V_{\mathrm{add},p^m}$ are now defined by

$$V_{\mathrm{add},|Q|^m}(S) = V_{\mathrm{sub},|Q|^m}^{-1}(V_{\mathrm{sub}}(S)) \cup (S \setminus V_{\mathrm{sub}}(S))$$

and

$$V_{\mathrm{add},p^m}(S) = V_{\mathrm{sub},p^m}^{-1}(V_{\mathrm{sub}}(S)) \cup (S \setminus V_{\mathrm{sub}}(S)).$$

Now, since both $V_{\mathrm{sub}}(p^m)$ and $V_{\mathrm{sub}}(|Q|^m)$ are close to $p_{\mathrm{core}}$, the distributions $V_{\mathrm{add},|Q|^m}(p^m)$ and $V_{\mathrm{add},p^m}(|Q|^m)$ are both close in TV-distance to the distribution of samples

$$V_{\mathrm{add},|Q|^m}(S_{\mathrm{core}}) \cup V_{\mathrm{add},p^m}(S_{\mathrm{core}}) \setminus S_{\mathrm{core}},$$

where $S_{\mathrm{core}} \sim p_{\mathrm{core}}$. In particular, the only difference between $V_{\mathrm{add},|Q|^m}(p^m)$ and $V_{\mathrm{add},p^m}(|Q|^m)$ can be understood as differences in the sampling of $S_{\mathrm{core}}$, as given $S_{\mathrm{core}}$, the distribution of additional samples is the same in both cases. Thus, $d_{\mathrm{TV}}(V_{\mathrm{add},|Q|^m}(p^m), V_{\mathrm{add},p^m}(|Q|^m)) \leq d_{\mathrm{TV}}(V_{\mathrm{sub}}(p^m), V_{\mathrm{sub}}(|Q|^m)) < \zeta$. $\qquad\square$

This intuition is made rigorous in the full proof of Theorem 6.1 in the appendix.

As a corollary of the above theorem, we can state a simple condition for a class $\mathcal{C}$ and a subtractive adversary $V_{\mathrm{sub}}$, that implies hardness for both adaptive additive and adaptive subtractive robust learning.

**Corollary 6.2.** *Let $\mathcal{C}$ be a class of distributions and $V_{\mathrm{sub}}$ be an adaptive subtractive adversary with a budget with* budget$^{\mathrm{sub}}(V_{\mathrm{sub}}) = \eta$. *If there are constants $0 < \gamma', \zeta < 1$, such that for every $m \in \mathbb{N}$, the adversary $V_{\mathrm{sub}}$ successfully $(2\alpha\eta + 2\gamma')$-confuses $\mathcal{C}$-generated samples of size $m$, then $\mathcal{C}$ is neither adaptively subtractive $\alpha$-robustly learnable, nor adaptively additive $\alpha$-robustly learnable.*

*Proof.* This corollary directly follows from Theorem 6.1 and Theorem 5.2. $\qquad\square$

**Corollary 6.3.** *For every $\alpha > 1$, the class $\mathcal{C}_g$ is not adaptively $\alpha$-robustly learnable.*

*Proof.* This corollary follows directly from Corollary 6.2 and Lemma 5.5. $\qquad\square$

While this already shows a separation between adaptive and oblivious additive robustness, before finally proving Theorem 4.1, we first need to show the existence of a *universal* adaptive additive adversary.

# 7. Universal Additive Adversaries

In this section, address the existence of universal adaptive additive adversaries. We know from Theorem 5.2 that if a single adaptive subtractive adversary $V_{\mathrm{sub}}$ successfully confuses $\mathcal{C}$-generated samples of all sizes, then $V_{\mathrm{sub}}$ is a universal adversary for $\mathcal{C}$. Moreover, we have seen in Theorem 6.1 that the existence of such a single subtractive adversary also implies the existence of a pair of adaptive additive adversaries that successfully confuses $\mathcal{C}$-generated examples and thus also shows that $\mathcal{C}$ is not adaptively additively learnable. However, these results do not yet show the existence of a universal adaptive additive adversary for $\mathcal{C}$. In the following theorem, we will show that the existence of a subtractive adversary $V_{\mathrm{sub}}$ that successfully confuses $\mathcal{C}$-generated samples also implies the existence of an adaptive additive universal adversary for $\mathcal{C}$, albeit one with a higher budget than $V_{\mathrm{sub}}$.

**Theorem 7.1.** *Let $\mathcal{C}$ be a class of distributions. Let $V_{\mathrm{sub}}$ be a adaptive subtractive adversary. Let $\zeta \in (0, 1)$ be a constant, $p \in \mathcal{C}$ a distribution and $Q$ a distribution over elements in $\mathcal{C}$ such that: $d_{\mathrm{TV}}(V_{\mathrm{sub}}(|Q|^m), V_{\mathrm{sub}}(p^m)) < \zeta$. Then for every $k \in \mathbb{N}$ there is an adaptive additive adversary $V_{\mathrm{add},k}$ with $d_{\mathrm{TV}}(V_{\mathrm{add},k}(|Q|^m), V_{\mathrm{add},k}(p^m)) < \zeta + \frac{1}{k+1}$. Furthermore, if $V_{\mathrm{sub}}$ has a fixed constant budget of $\eta < \frac{2}{k}$, then $V_{\mathrm{add},k}$ has a fixed constant budget of no more than $k\eta$.*

*Proof sketch.* Given a subtractive adversary $V_{\mathrm{sub}}$ as before, the additive adversary obtains new samples by first applying $V_{\mathrm{sub}}$ to obtain a subset $S' = V_{\mathrm{sub}}(S) \subset S$. We then again use the reverse mappings $V_{\mathrm{sub},p^m}^{-1}$ and $V_{\mathrm{sub},|Q|^m}^{-1}$ to obtain new sample points. However, now, in contrast to the previous theorem, the idea is not to apply $V_{\mathrm{sub},p^m}^{-1}$ or $V_{\mathrm{sub},|Q|^m}^{-1}$ just once for their respective distribution. Instead, the adversary makes use of both of these mappings a randomized number of times. That is, the adversary $V_{\mathrm{add},k}$ picks a number $u$ from $\{0, 1, \dots, k\}$ uniformly at random. Then, for every $i \in [k]$ it generates a sample $S_i'' = V_{\mathrm{sub},p^m}^{-1}(S')$ if $i \leq u$ and $S_i'' = V_{\mathrm{sub},|Q|^m}^{-1}(S')$ otherwise. All newly obtained samples are then concatenated with the original $S$ to produce the output sample

$$S'' = S' \cup (S \setminus S') \cup (S_1'' \setminus S') \cup \cdots \cup (S_k'' \setminus S').$$

Now consider the number of different subsamples within this concatenation that are generated by $V_{\mathrm{sub},p^m}^{-1}(S')$. This number is $u+1$ if the initial sample $S$ was generated by $S \sim p^m = V_{\mathrm{sub},p^m}^{-1}(S')$ and this number is $u$ if the initial sample $S$ was generated by $S \sim |Q|^m = V_{\mathrm{sub},|Q|^m}^{-1}(S')$. The resulting total variation distance $d_{\mathrm{TV}}(V_{\mathrm{add},k}(|Q|^m), V_{\mathrm{add},k}(p^m))$

is thus upper bounded by

$$
\begin{aligned}
&d_{\mathrm{TV}}(V_{\mathrm{add},k}(|Q|^m), V_{\mathrm{add},k}(p^m)) \\
&\quad \leq d_{\mathrm{TV}}(V_{\mathrm{sub}}(|Q|^m), V_{\mathrm{sub}}(p^m)) \\
&\qquad + d_{\mathrm{TV}}(U_{\{0,1,\dots,k\}}, U_{\{1,\dots,k,k+1\}}) \\
&\quad = \zeta + \frac{1}{k+1}.
\end{aligned}
$$

$\square$

The full proof can be found in the appendix.

As a result we obtain the following corollary.

**Corollary 7.2.** *Let $\mathcal{C}$ be a class of distributions and $V_{\mathrm{sub}}$ be an adaptive subtractive adversary with constant budget $\mathrm{budget}^{\mathrm{sub}}(V_{\mathrm{sub}}) = \eta$. If there are constants $0 < \gamma, \zeta < \frac{1}{2}$, such that for every $m \in \mathbb{N}$, $V_{\mathrm{sub}}$ successfully $(4\alpha\eta + 4\gamma', \zeta)$-confuses $\mathcal{C}$-generated samples of size $m$, then there is a universal additive $\alpha$-adversary $V_{\mathrm{add},2}$ for $\mathcal{C}$.*

*Proof.* This corollary directly follows from Theorem 7.1 and Theorem 5.2. $\square$

## 7.1. Proof of Theorem 4.1

We can now prove the main theorem of this paper, Theorem 4.1. A separation between the power of adaptive and oblivious additive adversaries follows as a corollary.

*Proof.* From Lemma 4.2 we know that $\mathcal{C}_g$ is realizably learnable with sample complexity function $m_{\mathcal{C}_g}^{\mathrm{re}}(\varepsilon, \delta) \leq \log\left(\frac{1}{\delta}\right) g\left(\frac{1}{\varepsilon}\right)$. From Lemma 5.5 and Corollary 6.2, we can infer that for every $\alpha \geq 1$ there is a universal subtractive adversary $V_{\mathrm{sub},\eta}$ with budget $\eta$, such that for every $m \in \mathbb{N}$, $V_{\mathrm{sub},\eta}$ is successfully $(4\alpha\eta + 4\gamma', \zeta)$-confusing $\mathcal{C}_g$ generated examples of size $m$. Finally, from Corollary 7.2, we get that the above implies that there is a adaptive additive adversary that has budget $2\eta$ and is a universal $\alpha$-adversary for $\mathcal{C}_g$. $\square$

Since it has been shown that classes that are realizably learnable are also learnable in the oblivious additive 3-robust case, this result shows a separation between learnability between adaptive additive and oblivious additive learnability.

**Corollary 7.3.** *There is a class $\mathcal{C}$ that is (obliviously) additive 3-robustly learnable, but for every $\alpha \geq 1$, $\mathcal{C}_g$ is not adaptively additive $\alpha$-robustly learnable.*

*Proof.* This result follows directly from Theorem 4.1 and Theorem 1.5 of Ben-David et al. (2023). $\square$

## Acknowledgments

We would like to thank Shai Ben-David for helpful discussions.

## Impact Statement

This paper presents work whose goal is to advance the field of Machine Learning. There are many potential societal consequences of our work, none which we feel must be specifically highlighted here.

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

# A. Proofs

## A.1. Proof of Lemma 5.3

*Proof.* By definition of $(\gamma, \zeta)$-confusion of $\mathcal{C}$-generated samples of size $m$, there are distributions $p \in \mathcal{C}$ and $|Q| \in \Delta(\mathcal{C})$, such that

- for every $q \in \mathrm{supp}(Q)$, we have $d_{\mathrm{TV}}(p, q) > \gamma$ and
- $d_{\mathrm{TV}}(V_1(|Q|^m), V_2(p^m)) < \zeta$.

Assume by way of contradiction that there is a learner $A$ such that for every $r \in \mathcal{C}$,

$$\mathbb{P}_{S \sim r^m} \left[ d_{\mathrm{TV}}(A(V_1(S)), r) > \frac{\gamma}{2} \right] < \frac{1}{2} - \frac{\zeta}{2}$$

and

$$\mathbb{P}_{S \sim r^m} \left[ d_{\mathrm{TV}}(A(V_2(S)), r) > \frac{\gamma}{2} \right] < \frac{1}{2} - \frac{\zeta}{2}.$$

In particular, this means that for $p \in \mathcal{C}$, we have

$$\mathbb{P}_{S \sim p^m} \left[ d_{\mathrm{TV}}(A(V_1(S)), p) \leq \frac{\gamma}{2} \right] \geq 1 - \left( \frac{1}{2} - \frac{\zeta}{2} \right) = \frac{1}{2} + \frac{\zeta}{2}$$

and

$$\mathbb{P}_{S \sim p^m} \left[ d_{\mathrm{TV}}(A(V_2(S)), p) \leq \frac{\gamma}{2} \right] \geq 1 - \left( \frac{1}{2} - \frac{\zeta}{2} \right) = \frac{1}{2} + \frac{\zeta}{2}. \tag{1}$$

We note that for any $p_1, p_2$ with $d_{\mathrm{TV}}(p_1, p_2) < d$ and any predicate $F$, we have

$$\mathbb{P}_{x \sim p_2}[F(x)] - d \leq \mathbb{P}_{x \sim p_1}[F(x)] \leq \mathbb{P}_{x \sim p_2}[F(x)] + d.$$

Thus, for meta-distribution $Q$ with $d_{\mathrm{TV}}(V_1(|Q|^m), V_2(p^m))] < \zeta$ we have,

$$\mathbb{P}_{S \sim |Q|^m} \left[ d_{\mathrm{TV}}(A(V_1(S)), p) \leq \frac{\gamma}{2} \right] \geq \mathbb{P}_{S \sim p^m} \left[ d_{\mathrm{TV}}(A(V_2(S)), p) \leq \frac{\gamma}{2} \right] - \zeta \geq_{(1)} \frac{1}{2} + \frac{\zeta}{2} - \zeta = \frac{1}{2} - \frac{\zeta}{2}. \tag{2}$$

Furthermore, we have

$$\max_{q \in \mathrm{supp}(Q)} \mathbb{P}_{S \sim q^m} \left[ d_{\mathrm{TV}}(A(V_1(S)), p) \leq \frac{\gamma}{2} \right]$$

$$\geq \qquad \mathbb{P}_{q \sim Q} \mathbb{P}_{S \sim q^m} \left[ d_{\mathrm{TV}}(A(V_1(S)), p) \leq \frac{\gamma}{2} \right]$$

$$=_{\text{Definition of } |Q|^m} \qquad \mathbb{P}_{S \sim |Q|^m} \left[ d_{\mathrm{TV}}(A(V_1(S)), p) \leq \frac{\gamma}{2} \right]$$

$$\geq_{(2)} \qquad \frac{1}{2} - \frac{\zeta}{2}. \tag{3}$$

Let

$$q_{\max} = \arg \max_{q \in \mathrm{supp}(Q)} \mathbb{P}_{S \sim q^m} \left[ d_{\mathrm{TV}}(A(V_1(S)), p) \leq \frac{\gamma}{2} \right].$$

Recall that, for every $q \in \mathrm{supp}(Q)$, we have $d_{\mathrm{TV}}(p, q) > \gamma$. Thus triangle inequality yields

$$d_{\mathrm{TV}}(A(V_1(S)), q_{\max}) + d_{\mathrm{TV}}(A(V_1(S)), p) > \gamma.$$

Thus, $d_{\mathrm{TV}}(A(V_1(S), p)) \leq \frac{\gamma}{2}$ implies $d_{\mathrm{TV}}(A(V_1(S), q_{\max})) > \frac{\gamma}{2}$, yielding,

$$\mathbb{P}_{S \sim q_{\max}} \left[ d_{\mathrm{TV}}(A(V_1(S)), q_{\max}) > \frac{\gamma}{2} \right]$$

$$\geq \qquad \mathbb{P}_{S \sim q_{\max}} \left[ d_{\mathrm{TV}}(A(V_1(S)), p) \leq \frac{\gamma}{2} \right]$$

$$\geq_{(3)} \qquad \frac{1}{2} - \frac{\zeta}{2}.$$

This contradicts our assumption on $A$, which proves the claim. $\qquad \square$

### A.2. Proof of Theorem 5.2

*Proof.* Assume by way of contradiction that there was a successful $\alpha$-robust learner $A$ with sample complexity $m_{\mathcal{C}}$ for $\mathcal{C}$ with respect to $\mathcal{V} \supset \{V_1, V_2\}$.

Let

$$\delta = \min \left\{ \frac{1 - \zeta}{2}, \delta' \right\}$$

and

$$\varepsilon = \min\{\gamma', \varepsilon' - \alpha \max\{\eta_1, \eta_2\}\}.$$

Furthermore, let $m = m_{\mathcal{C}}^{\mathcal{V}, \alpha}(\varepsilon, \delta)$.

According to the assumptions of the theorem, we know that the pair $(V_1, V_2)$ successfully $(\gamma, \zeta)$-confuses $\mathcal{C}$-generated samples of size $m$ with

$$\gamma = 2\alpha \cdot \max\{\eta_1, \eta_2\} + 2\gamma.$$

Now consider

$$\alpha \cdot \eta_1 + \varepsilon = \alpha \cdot \eta_1 + \gamma'$$
$$\leq \frac{\gamma}{2}.$$

With the same argument, we have $\alpha \cdot \eta_2 + \varepsilon \leq \frac{\gamma}{2}$. Now using Lemma 5.3, we can infer that there is a distribution $r \in \mathcal{C}$ such that either

$$\mathbb{P}_{S \sim r^m} \left[ d_{\mathrm{TV}}(A(V_1(S)), r) > \alpha \cdot \eta_1 + \varepsilon \right]$$
$$\geq \mathbb{P}_{S \sim r^m} \left[ d_{\mathrm{TV}}(A(V_1(S)), r) > \frac{\gamma}{2} \right]$$
$$\geq \frac{1}{2} - \frac{\zeta}{2} = \delta.$$

or

$$\mathbb{P}_{S \sim r^m} \left[ d_{\mathrm{TV}}(A(V_2(S)), r) > \alpha \cdot \eta_2 + \varepsilon \right]$$
$$\geq \mathbb{P}_{S \sim r^m} \left[ d_{\mathrm{TV}}(A(V_2(S)), r) > \frac{\gamma}{2} \right]$$
$$\geq \frac{1}{2} - \frac{\zeta}{2} = \delta.$$

This is a contradiction to the assumption that $A$ is a $\alpha$-robust learner of $\mathcal{C}$ with respect to $\mathcal{V}$ with sample complexity $m_{\mathcal{C}}^{\mathcal{V}, \alpha}$. Furthermore, if $V_1 = V_2$, then $V_1$ is a universal $\alpha$-adversary.

$\square$

### A.3. Proof of Lemma 5.5

*Proof.* We will start by making observations for the adversary $V_{\mathrm{sub}, \eta}$ for arbitrary $\eta > 0$, and later discuss how to choose $\eta$ for a given $\alpha$. Similarly, we first start by making observations for general $n, i, j, k \in \mathbb{N}$ and then discuss appropriate choices for these numbers.

We first note that for $p = p_{i,j,k}$ with $|B_i| = 2^{2^n}$ and $D_{i,n,j,k} = \{p_{i',j,k} : B_{i'} \subset B_i : |B_{i'}| = 2^n\}$, we have that for every $q \in D_{i,n,j,k}$

$$d_{\mathrm{TV}}(p, q) \geq \left( \frac{1}{j} - \frac{1}{k} \right) d_{\mathrm{TV}}(U_{B_i \times 2j}, U_{B_{i'} \times 2j})$$
$$\geq \frac{1}{2} \left( \frac{1}{j} - \frac{1}{k} \right).$$

Furthermore, consider $Q = U_{D_{i,n,j,k}}$.

We note that the distributions $V_{\text{sub},\eta}(p^m)$ and $V_{\text{sub},\eta}(|Q|^m)$ are both distributions over multisets $S' = \text{constants}(S') \cup \text{odds}(S') \cup \text{indicators}(S')$. We note that for any two samples $S'_a \in \mathcal{X}^*$ and $S'_b \in \mathcal{X}^*$ by definitions of constants, odds, ind, we have

$$S'_a \cap S'_b =$$
$$(\text{constants}(S'_a) \cap \text{constants}(S'_b)) \cup (\text{odds}(S'_a) \cap \text{odds}(S'_b)) \cup (\text{ind}(S'_a) \cap \text{ind}(S'_b)).$$

Where each of the three sets $(\text{constants}(S'_a) \cap \text{constants}(S'_b)), (\text{odds}(S'_a) \cap \text{odds}(S'_b)), (\text{ind}(S'_a) \cap \text{ind}(S'_b))$ are pairwise disjoint. We can thus write the total variation distance between $V_{\text{sub},\eta}(p^m)$ and $V_{\text{sub},\eta}(|Q|^m)$ as

$$d_{\text{TV}}(V_{\text{sub},\eta}(p^m), V_{\text{sub},\eta}(|Q|^m))$$
$$= d_{\text{TV}}(\text{constants}(V_{\text{sub},\eta}(p^m)), \text{constants}(V_{\text{sub},\eta}(|Q|^m)))$$
$$+ d_{\text{TV}}(\text{odds}(V_{\text{sub},\eta}(p^m)), \text{odds}(V_{\text{sub},\eta}(|Q|^m)))$$
$$+ d_{\text{TV}}(\text{ind}(V_{\text{sub},\eta}(p^m)), \text{ind}(V_{\text{sub},\eta}(|Q|^m)))$$

For a distribution $P \in \Delta(\mathcal{X}^*)$ over sets, we define the distribution $\text{count}(P) \in \Delta(\mathbb{N})$ by

$$\forall B \subset \mathbb{N} : \text{count}(P)(B) = \mathbb{P}_{S \sim P}[|S| \in B].$$

We note that the distributions of the count of each of the subsets are the same for both $V_{\text{sub},\eta}(p^m)$ and $V_{\text{sub},\eta}(|Q|^m)$. That is,

$$d_{\text{TV}}(\text{count}(\text{constants}(V_{\text{sub},\eta}(p^m))), \text{count}(\text{constants}(V_{\text{sub},\eta}(|Q|^m)))) = 0,$$
$$d_{\text{TV}}(\text{count}(\text{odds}(V_{\text{sub},\eta}(p^m))), \text{count}(\text{odds}(V_{\text{sub},\eta}(|Q|^m)))) = 0,$$

and

$$d_{\text{TV}}(\text{count}(\text{indicators}(V_{\text{sub},\eta}(p^m))), \text{count}(\text{indicators}(V_{\text{sub},\eta}(|Q|^m))|) = 0.$$

Furthermore, for two different samples $S_a$ and $S_b$, $\text{constants}(S_a) = \text{constants}(S_b)$ if and only if $|\text{constants}(S_a)| = |\text{constants}(S_b)|$. Thus,

$$d_{\text{TV}}(\text{constants}(V_{\text{sub},\eta}(p^m)), \text{constants}(V_{\text{sub},\eta}(|Q|^m))) = 0.$$

Furthermore,

$$\mathbb{P}_{S' \sim V_{\text{sub},\eta}(p^m)}[\text{odds}(S')||\text{odds}(S')| = l \text{ and there are no repeated elements in } \text{odds}(S')] =$$
$$\mathbb{P}_{S' \sim V_{\text{sub},\eta}(|Q|^m)}[\text{odds}(S')||\text{odds}(S')| = l \text{ and there are no repeated elements in } \text{odds}(S')],$$

since both $|Q|^m$ and $p^m$ give equal weights to all subsets of $B_i$ the same size. Lastly, we note that while the indicator of samples from $p$ and samples from $Q$ differ, the adversary $V_{\text{sub},\eta}$ deletes all these elements, if $|\text{indicators}(S)|$ does not exceed the budget-count.

Taking all of these observations together, we can bound the total variation distance in terms of repeating elements in $\text{odds}(S)$ and the probability that $|\text{ind}(S)|$ exceeds the budget of the adversary.

$$d_{\text{TV}}(V_{\text{sub},\eta}(p^m), V_{\text{sub},\eta}(|Q|^m))$$
$$\leq d_{\text{TV}}(\text{constants}(V_{\text{sub},\eta}(p^m)), \text{constants}(V_{\text{sub},\eta}(|Q|^m)))$$
$$+ d_{\text{TV}}(\text{odds}(V_{\text{sub},\eta}(p^m)), \text{odds}(V_{\text{sub},\eta}(|Q|^m)))$$
$$+ d_{\text{TV}}(\text{ind}(V_{\text{sub},\eta}(p^m)), \text{ind}(V_{\text{sub},\eta}(|Q|^m)))$$
$$\leq \mathbb{P}_{S \sim p^m}[\text{odds}(S) \text{ contains repeated elements}]$$
$$+ \mathbb{P}_{S \sim |Q|^m}[\text{odds}(S) \text{ contains repeated elements}]$$
$$+ \mathbb{P}_{S \sim p^m}[|\text{ind}(S)| > \eta|S|]$$
$$+ \mathbb{P}_{S \sim |Q|^m}[|\text{ind}(S)| > \eta|S|].$$

The first two terms can be bounded via a birthday paradox:

$$\mathbb{P}_{S \sim p^m}[\text{odds}(S) \text{ contains repeated elements}] + \mathbb{P}_{S \sim |Q|^m}[\text{odds}(S) \text{ contains repeated elements}] \leq 2 \cdot \left(1 - \left(1 - \frac{m}{2^n}\right)^m\right)$$

The last two terms can each be upper bounded by using Markov's inequality:

$$\mathbb{P}_{S \sim p^m}[|\text{ind}(S)| > \eta|S|] + \mathbb{P}_{S \sim |Q|^m}[|\text{ind}(S)| > \eta|S|] \leq 2\frac{\frac{m}{k}}{\eta m} = \frac{2}{k\eta}$$

Since we are considering distributions in $\mathcal{C}_g$, we note that the distributions $D_{i,n,j,k}$ need to be of the form $D_{i,n,j,g(j)}$. We now want to argue that for appropriate choices of $j$ and $\eta$ (both independent of $m$), as well as for an appropriate choice of $n$ given $m$, the inequalities of the theorem are satisfied.

First, we note that since $g$ grows faster than linear, it is possible to pick $j$ in such a way that it satisfies the inequality

$$g(j) \geq \max\{1024\alpha j\}.$$

Given such $j$, we then pick $\eta = \frac{32}{g(j)}$ and $\gamma' = \frac{\alpha}{g(j)}$. This ensures, that for every $n \in \mathbb{N}$ and every $q \in D_{i,n \cdot j, g(j)}$, we get

$$
\begin{aligned}
d_{\text{TV}}(p, q) &\geq \frac{1}{2}\left(\frac{1}{j} - \frac{1}{g(j)}\right) \\
&\geq \frac{1}{2}\left(\frac{1024\alpha}{g(j)} - \frac{1}{g(j)}\right) \\
&> \frac{256\alpha}{g(j)} \\
&= 4\alpha\eta + 4\gamma'.
\end{aligned}
$$

Then, for every $m \geq 1$, if we choose $n \geq \frac{m}{1 - \left(1 - \frac{1}{32}\right)^{1/m}}$, we get

$$
\begin{aligned}
&d_{\text{TV}}(V_{\text{sub},\eta}(p^m), V_{\text{sub},\eta}(|Q|^m)) \\
&\leq \frac{2}{g(j) \cdot \frac{32}{g(j)}} + 2\left(1 - \left(1 - \frac{m}{2^n}\right)^m\right) \\
&\leq \frac{1}{16} + 2\left(1 - \left(1 - \frac{m}{n}\right)^m\right) \\
&\leq 2\exp\left(-2m\left(\frac{c-1}{g(j)}\right)^2\right) + 2\left(1 - \left(1 - \frac{m}{\frac{m}{\left(1 - \left(1 - \frac{1}{32}\right)^{1/m}\right)}}\right)^m\right) \\
&\leq \frac{1}{16} + 2 \cdot \frac{1}{32} \\
&\leq \frac{1}{8}.
\end{aligned}
$$

$\square$

### A.4. Proof of Theorem 6.1

*Proof.* While in the main part of the paper, we often use the same notation for a random variable $S \sim p^m$ and a specific sample set $S \in \mathcal{X}^*$, in order to make this proof more formal, we will now distinguish between random variables $S \sim p^m$, which we keep writing with capitalized notation and specific sample set $s \in \mathcal{X}^*$ for which we use non-capitalized notation.

We note, that since adversaries $V$ are in general randomized, for every $s \in \mathcal{X}^*$, $V(s)$ is a random variable with samples in $s' \in \mathcal{X}^*$. For every adversary $V$, let us denote the distribution of random variable $V(s)$ by $p_{V(s)}$.

We note, that for a random variable $S \sim r^m$, the distribution of the random variable $V(S)$ defined for every $B \subset \mathcal{X}^*$ by

$$p_{V,r}(B) = \int_{s' \in B} \int_{s \in \mathcal{X}^*} dp_{V(s)}(s') dr(s).$$

Accordingly, we have

$$dp_{V,r}(s') = \int_{s \in \mathcal{X}^*} dp_{V(s)}(s') dr(s).$$

We now want to formally define the reverse mapping $f_{V,r,s'}$ that, when given an input sample $S' \sim V(S)$ with $S \sim r^m$, outputs a sample $S'' \sim r^m$. That is, roughly, $f_{V,r,s'}(s) = \mathbb{P}_{S \sim r^m}[S = s | V(S) = s']$.

For a distribution $r$ over $\mathcal{X}^*$ and an adversary $V$, let us consider the random function $f_{V,r}^{-1}$ that takes as input a sample $s' \in \mathcal{X}^*$ and outputs an element of $\mathcal{X}^*$ according to the probability distribution $p_{V^{-1},r,s'}$ which for every $B \subset \mathcal{X}^*$ is defined by

$$p_{V^{-1},r,s'}(B) = \frac{\int_{s \in B} dp_{V(s)}(s') dr(s)}{\int_{S \in \mathcal{X}^m} dp_{V(s)}(s') dr(s)}.$$

Similarly,

$$dp_{V^{-1},r,s'}(s'') = \frac{dp_{V(s'')}(s') dr(s'')}{\int_{s \in \mathcal{X}^m} dp_{V(s)}(s') dr(s)}.$$

Thus, if we consider the random variable $S'' = f_{V,r}^{-1}(V(S))$ for some $S \sim r$, we get $S'' \sim r$, as desired:

$$\mathbb{P}_{S \sim r}[f_{V,r}^{-1}(V(S)) \in B''] = \int_{s'' \in B''} \int_{s \in \mathcal{X}^*} \int_{s' \in \mathcal{X}^*} dp_{V^{-1},r,s'}(s'') dp_{V(s)}(s') dr(s)$$

$$= \int_{s'' \in B''} \int_{s \in \mathcal{X}^*} \int_{s' \in \mathcal{X}^*} \frac{dp_{V(s'')}(s') dr(s'')}{\int_{s''' \in \mathcal{X}^m} dp_{V(s''')}(s') dr(s''')} dp_{V(s)}(s') dr(s)$$

$$= \int_{s'' \in B''} \int_{s' \in \mathcal{X}^*} \frac{\int_{s \in \mathcal{X}^*} dp_{V(s)}(s') dr(s) dp_{V(s'')}(s') dr(s'')}{\int_{s''' \in \mathcal{X}^m} dp_{V(s''')}(s') dr(s''')}$$

$$= \int_{s'' \in B''} \int_{s' \in \mathcal{X}^*} dp_{V(s'')}(s') dr(s'') \frac{\int_{s \in \mathcal{X}^*} dp_{V(s)}(s') dr(s)}{\int_{s''' \in \mathcal{X}^m} dp_{V(s''')}(s') dr(s''')}$$

$$= \int_{s'' \in B''} \int_{s' \in \mathcal{X}^*} dp_{V(s'')}(s') dr(s'')$$

$$= \int_{s'' \in B''} dr(s'')$$

$$= \mathbb{P}_{S \sim r}[S \in B''].$$

We note, that by the same argument, we can factorize $r$ in the following way:

$$r(B) = \int_{s \in B} \int_{s' \in \mathcal{X}^*} dp_{V,r}(s') dp_{V^{-1},r,s'}(s).$$

Now consider a subtractive adversary $V_{\text{sub}}$. Since it is subtractive adversary, we know that for every $s \in \mathcal{X}^m$ and every $s' \in \text{supp}(p_{V_{\text{sub}}(s)} \subset \bigcup_{i=(1-b)m}^{m} X^i$, we have $s' \subset s$.

In particular, this means that for every $s \in \mathcal{X}^m$ and every $s' \in \text{supp}(p_{V_{\text{sub}}(s)})$, we have $s = s' \cup (s \setminus S')$. Now let $f_{V^{-1},r}^{\setminus}(s) = f_{V^{-1},r}(s) \setminus s$ with the corresponding probability measure.

$$p_{V^{-1},r,s}^{\setminus}(B) = p_{V^{-1},r,s}(B'),$$

where $B' = \{s \cup s' : s' \in B\}$.

Now consider the (randomized) additive adversary $V_{\text{add},r}$, defined by:

$$V_{\text{add},r}(s) = S \cup f_{V_{\text{sub}}^{-1},r}^{\backslash}(V_{\text{sub}}(s))).$$

Thus for the corresponding probability measure for the random variable $V_{\text{add},r}(s)$ is defined by

$$p_{V_{\text{add},r}(s)}(B''') = \int_{s''' \in B'''} \int_{s'' \in \mathcal{X}^*} \int_{s' \in X^*} dp_{V_{\text{sub}}(s)}(s') dp_{V_{\text{sub},r,s'}^{-1}}(s'') \mathbb{1}[s''' = s' \cup (s \setminus s') \cup (s'' \setminus s')]$$

Furthermore, for every probability distribution $q$ we have.

$$\begin{aligned}
&\mathrm{P}_{S \sim q}[V_{\text{add},r}(S) \in B'''] \\
&= \int_{s''' \in B'''} \int_{s \in \mathcal{X}^*} dp_{V_{\text{add},r}(s)}(s''') dq(s) \\
&= \int_{s''' \in B'''} \int_{s \in \mathcal{X}^*} \int_{s' \in X^*} \int_{s' \in X^*} dp_{V_{\text{sub}}(s)}(s') dp_{V_{\text{sub},r,s'}^{-1}}(s'') \mathbb{1}[s''' = s' \cup (s \setminus s') \cup (s'' \setminus s')] dq(s) \\
&= \int_{s''' \in B'''} \int_{s \in \mathcal{X}^*} \int_{s' \in X^*} \int_{s' \in X^*} \mathbb{1}[s''' = s' \cup (s \setminus s') \cup (s'' \setminus s')] dp_{V_{\text{sub}},q}(s') dp_{V^{-1},q,s'}(s) dp_{V^{-1},r,s'}(s'')
\end{aligned}$$

Now consider the additive adversaries $V_{\text{add},p^m}$ and $V_{\text{add},|Q|^m}$.

$$\begin{aligned}
&d_{\text{TV}}(V_{\text{add},p^m}(|Q|^m), V_{\text{add},|Q|^m}(p^m)) \\
&= \frac{1}{2} \int_{s''' \in X^*} \left| \int_{s \in \mathcal{X}^m} dp_{V_{\text{add},p^m}(s''')} d|Q|^m(s) - \int_{s \in \mathcal{X}^m} dp_{V_{\text{add},p^m}(s''')} d|Q|^m(s) \right| \\
&= \frac{1}{2} \int_{s''' \in X^*} \left| \int_{s' \in \mathcal{X}^*} \int_{s \in \mathcal{X}^m} \int_{s'' \in \mathcal{X}^*} \mathbb{1}[s''' = s' \cup (s \setminus s') \cup (s'' \setminus s')] dp_{V_{\text{sub}},|Q|^m}(s') dp_{V^{-1},|Q|^m,s'}(s) dp_{V^{-1},p^m,s'}(s'') \right. \\
&\quad \left. - \int_{s' \in \mathcal{X}^*} \int_{s \in \mathcal{X}^*} \int_{s''' \in \mathcal{X}^*} \mathbb{1}[s''' = s' \cup (s \setminus s') \cup (s'' \setminus s')] dp_{V_{\text{sub}},p^m}(s') dp_{V^{-1},|Q|^m,s'}(s) dp_{V^{-1},p^m,s'}(s'') \right| \\
&= \frac{1}{2} \int_{s''' \in X^*} \left| \int_{s' \in \mathcal{X}^*} |dp_{V_{\text{sub}},|Q|^m}(s') - dp_{V_{\text{sub}},p^m}(s'))| \right. \\
&\quad \left. \cdot \left| \int_{s \in \mathcal{X}^m} \int_{s'' \in \mathcal{X}^*} \mathbb{1}[s''' = s' \cup (s \setminus s') \cup (s'' \setminus s')] dp_{V^{-1},|Q|^m,s'}(s) dp_{V^{-1},p^m,s'}(s'')' \right| \right. \\
&= \frac{1}{2} \left| \int_{s' \in \mathcal{X}^*} |dp_{V_{\text{sub}},|Q|^m}(s') - dp_{V_{\text{sub}},p^m}(s'))| \right. \\
&\quad \left. \cdot \int_{s''' \in X^*} \left| \int_{s \in \mathcal{X}^m} \int_{s'' \in \mathcal{X}^*} \mathbb{1}[s''' = s' \cup (s \setminus s') \cup (s'' \setminus s')] dp_{V^{-1},|Q|^m,s'}(s) dp_{V^{-1},p^m,s'}(s'') \right| \right| \\
&\leq \frac{1}{2} \left| \int_{s' \in \mathcal{X}^*} |dp_{V_{\text{sub}},|Q|^m}(s') - dp_{V_{\text{sub}},p^m}(s'))| \right| \\
&= d_{\text{TV}}(p_{V_{\text{sub}},p^m}, p_{V_{\text{sub}},|Q|^m}) \leq \zeta
\end{aligned}$$

where we get the second to last step by noticing that the last three integrals are a conditional probability distribution of the additive adversary outputting $s'''$, conditioned on the subtractive adversary outputting $s'$. As such, these integrals equate to 1.

Furthermore, we note that if $V_{\text{sub}}$ has a fixed constant budget with $\eta m - 1 < m \cdot \text{budget}^{\text{sub}}(V_{\text{sub}}, m) \leq \eta m$, then we have

$$
\begin{aligned}
\text{budget}^{\text{add}}(V_{\text{add},r}, m) &= \sup_{s \in \mathcal{X}^m} \frac{|V_{\text{add},r}(s)| - |s|}{|s|} \\
&\leq \sup_{s \in \mathcal{X}^m} \sup_{s':s' \in \text{supp}(p_{V(s)})} \sup_{s'':s'' \in \text{supp}\left(p_{V_{\text{sub},r}^{-1}(s')}\right)} \frac{|s \setminus s'| + |s''| - |s|}{|s|} \\
&\leq \max \left\{ \frac{\eta m - 1 + (m - (\eta m - 1))\frac{1}{1-\eta} - m}{m}, \frac{\eta m + (m - \eta m)\frac{1}{1-\eta} - m}{m} \right\} \\
&\leq \max \left\{ \eta - 1 + \frac{\eta + m - \eta m}{m(1-\eta)}, \eta \right\} \\
&\leq \max \left\{ \eta + \frac{\eta}{m(1-\eta)}, \eta \right\} \\
&\leq \eta + \frac{\eta}{m - \eta m}.
\end{aligned}
$$

In particular, this means, that

$$
m \cdot \text{budget}^{\text{add}}(V_{\text{add},r}, m) \leq \eta m + \frac{\eta}{(1-\eta)}.
$$

We note that $\frac{\eta}{1-\eta}$ is strictly monotonically increasing in $\eta$. Thus, for $\eta < \frac{1}{2}$ we thus get,

$$
m \cdot \text{budget}^{\text{add}}(V_{\text{add},r}, m) < \eta m + \frac{\frac{1}{2}}{\frac{1}{2}} = \eta m + 1.
$$

Thus, $\text{budget}^{\text{add}}(V_{\text{add},r}) \leq \eta$.

$\square$

## A.5. Proof of Theorem 7.1

*Proof.* Let $u$ be a random variable that is uniformly distributed over $[k+1] = \{0, \ldots, k\}$. Now let $V_{\text{add},k}$ be defined by the probability distribution

$$
\begin{aligned}
dp_{V_{\text{add},k(s)}}(s'') = &\int_{s' \in \mathcal{X}^*} dp_{V_{\text{sub}(s)}}(s') \\
&\cdot \frac{1}{k+1} \sum_{u=0}^{k} \int_{s_1 \in \mathcal{X}^*} \cdots \int_{s_k \in \mathcal{X}^*} \left( \Pi_{i=0}^{k}(\mathbb{1}[u \geq i]dp_{V_{\text{sub}}^{-1}, |\mathcal{Q}|^m, s'}(s_i) + \mathbb{1}[u < i]dp_{V_{\text{sub}}^{-1}, p^m, s'}(s_i)) \right) \\
&\cdot \mathbb{1}\left[ s'' = s' \cup (s \setminus s') \cup \left( \bigcup_{i=1}^{k}(s_i \setminus s') \right) \right].
\end{aligned}
$$

We now note that

$$\int_{s\in\mathcal{X}^m} dp_{V_{\mathrm{add},k(s)}}(s'')dp^m(s) = \int_{s\in\mathcal{X}^m}\int_{s'\in\mathcal{X}^*} dp_{V_{\mathrm{sub}}(s)}(s')$$

$$\cdot\frac{1}{k+1}\sum_{u=0}^{k}\int_{S_1\in\mathcal{X}^*}\cdots\int_{S_k\in\mathcal{X}^*}\left(\Pi_{i=0}^{k}(\mathbb{1}[u\geq i]dp_{V_{\mathrm{sub}}^{-1},|\mathcal{Q}|^m,S'}(S_i)+\mathbb{1}[u<i]dp_{V_{\mathrm{sub}}^{-1},p^m,S'}(S_i))\right)$$

$$\cdot\mathbb{1}\left[s''=s'\cup(s\setminus s')\cup\left(\bigcup_{i=1}^{k}(s_i\setminus s')\right)\right]$$

$$=\int_{s'\in\mathcal{X}^*} dp_{V_{\mathrm{sub},p^m}}(s')\int_{s\in\mathcal{X}^*} dp_{V_{\mathrm{sub},p^m,s'}}(s)$$

$$\cdot\frac{1}{k+1}\sum_{u=0}^{k}\int_{S_1\in\mathcal{X}^*}\cdots\int_{S_k\in\mathcal{X}^*}\left(\Pi_{i=0}^{k}(\mathbb{1}[u\geq i]dp_{V_{\mathrm{sub}}^{-1},|\mathcal{Q}|^m,s'}(s_i)+\mathbb{1}[u<i]dp_{V_{\mathrm{sub}}^{-1},p^m,s'}(s_i))\right)$$

$$\cdot\mathbb{1}\left[s''=s'\cup(s\setminus s')\cup\left(\bigcup_{i=1}^{k}(s_i\setminus s')\right)\right].$$

Similarly,

$$\int_{s\in\mathcal{X}^m} dp_{V_{\mathrm{add},k(s)}}(s'')dp^m(s) = \int_{s'\in\mathcal{X}^*} dp_{V_{\mathrm{sub},|Q|^m}}(s')\int_{s\in\mathcal{X}^*} dp_{V_{\mathrm{sub},|Q|^m,s'}}(s)$$

$$\cdot\frac{1}{k+1}\sum_{u=0}^{k}\int_{s_1\in\mathcal{X}^*}\cdots\int_{s_k\in\mathcal{X}^*}\left(\Pi_{i=0}^{k}(\mathbb{1}[u\geq i]dp_{V_{\mathrm{sub}}^{-1},|\mathcal{Q}|^m,s'}(s_i)+\mathbb{1}[u<i]dp_{V_{\mathrm{sub}}^{-1},p^m,s'}(S_i))\right)$$

$$\cdot\mathbb{1}\left[s''=s'\cup(s\setminus s')\cup\left(\bigcup_{i=1}^{k}(s_i\setminus s')\right).\right]$$

We now note that

$$\frac{1}{2}\int_{s'\in\mathcal{X}^*}\left|dp_{V_{\mathrm{sub},p^m}(s)}(s')-\int dp_{V_{\mathrm{sub}}(s),|Q|^m}(s')\right|\leq\zeta.$$

and

$$\frac{1}{k+1}\sum_{u=0}^{k}\int_{s\in\mathcal{X}^*} dp_{V_{\mathrm{sub},|Q|^m,s'}}(s)\int_{s_1\in\mathcal{X}^*}\cdots\int_{s_k\in\mathcal{X}^*}\left(\Pi_{i=0}^{k}(\mathbb{1}[u\geq i]dp_{V_{\mathrm{sub}}^{-1},|\mathcal{Q}|^m,s'}(s_i)+\mathbb{1}[u<i]dp_{V_{\mathrm{sub}}^{-1},p^m,s'}(s_i))\right)$$

$$\cdot\mathbb{1}\left[s''=s'\cup(s\setminus s')\cup\left(\bigcup_{i=1}^{k}(s_i\setminus s')\right).\right]$$

$$-\frac{1}{k+1}\sum_{u=0}^{k}\int_{s\in\mathcal{X}^*} dp_{V_{\mathrm{sub},p^m,s'}}(S)\int_{s_1\in\mathcal{X}^*}\cdots\int_{s_k\in\mathcal{X}^*}\left(\Pi_{i=0}^{k}(\mathbb{1}[u\geq i]dp_{V_{\mathrm{sub}}^{-1},|\mathcal{Q}|^m,s'}(s_i)+\mathbb{1}[u<i]dp_{V_{\mathrm{sub}}^{-1},p^m,s'}(s_i))\right)$$

$$\cdot\mathbb{1}\left[s''=s'\cup(s\setminus s')\cup\left(\bigcup_{i=1}^{k}(s_i\setminus s')\right).\right]$$

$$=\frac{1}{k+1}\int_{s\in\mathcal{X}^*} dp_{V_{\mathrm{sub},|Q|^m,s'}}(s)\int_{s_1\in\mathcal{X}^*}\cdots\int_{s_k\in\mathcal{X}^*}\left(\Pi_{i=0}^{k}(dp_{V_{\mathrm{sub}}^{-1},|\mathcal{Q}|^m,s'}(s_i))\cdot\mathbb{1}\left[s''=s'\cup(s\setminus s')\cup\left(\bigcup_{i=1}^{k}(s_i\setminus s')\right).\right]\right.$$

$$-\frac{1}{k+1}\int_{s\in\mathcal{X}^*} dp_{V_{\mathrm{sub},p^m,s'}}(S)\int_{S_1\in\mathcal{X}^*}\cdots\int_{S_k\in\mathcal{X}^*}\left(\Pi_{i=0}^{k}(dp_{V_{\mathrm{sub}}^{-1},p^m,s'}(s_i))\cdot\mathbb{1}\left[s''=s'\cup(s\setminus s')\cup\left(\bigcup_{i=1}^{k}(s_i\setminus s')\right).\right]\right.$$

$$\leq\frac{1}{k+1}.$$

This means that

$$d_{\text{TV}}(V_{\text{add},k}(p^m), V_{\text{add},k}(|Q|^m)) = \frac{1}{2} \int_{s'' \in \mathcal{X}^*} \left| \left( \int dp_{V_{\text{add},k}(s)}(s'') dp^m(s) - \int dp_{V_{\text{add},k}(s)}(s'') d|Q|^m(s) \right) \right|$$

$$\leq \zeta + (1 - \zeta) \frac{1}{k+1}.$$

Furthermore, we note that if $V_{\text{sub}}$ has a fixed constant budget with $\eta m - 1 < m \cdot \text{budget}^{\text{sub}}(V_{\text{sub}}, m) \leq \eta m$, then we have

$$\text{budget}^{\text{add}}(V_{\text{add},k}, m) = \sup_{s \in \mathcal{X}^m} \frac{|V_{\text{add},k}(s)| - |s|}{|s|}$$

$$\leq \max\{ \frac{(\eta m - 1) + (m - (\eta m - 1)) + k\left((m - (\eta m - 1))\frac{1}{1-\eta} - (m - (\eta m - 1))\right) - m}{m},$$

$$\frac{\eta m + (m - \eta m) + k(m - \eta m)\left(\frac{1}{1-\eta} - 1\right) - m}{m}\}$$

$$\leq \max\left\{ k\eta + \frac{k\eta}{(1-\eta)m}, k\eta \right\} \leq k\eta + \frac{k\eta}{(1-\eta)m}$$

In particular, this means, that

$$m \cdot \text{budget}^{\text{add}}(V_{\text{add},r}, m) \leq k\eta m + \frac{k\eta}{(1-\eta)}.$$

We note that $\frac{\eta}{1-\eta}$ is strictly monotonically increasing in $\eta$. Thus, for $\eta < \frac{1}{2k}$ we thus get,

$$m \cdot \text{budget}^{\text{add}}(V_{\text{add},r}, m) < k\eta m + \frac{1/2}{1 - 1/2k} < \eta m + 1.$$

Thus, $\text{budget}^{\text{add}}(V_{\text{add},r}) \leq \eta k$.

$\square$

## B. Additional Example for Usefulness of Lemma 5.4

In this subsection we give a short illustration of why the lemmas in Section 5 can be helpful. We give a known example for the hardness of PAC learning of distributions, which also fulfills the indistinguishability condition of Lemma 5.4.

*Example* B.1. Let $\mathcal{X} = \mathbb{N}$. Let $\zeta \in (0,1)$. Furthermore, let $p = U_B$ for some set $B \subset \mathbb{N}$ with $|B| = \frac{2^m m}{1 - (1-\zeta)^{1/m}}$ and let $\mathcal{C} = \{U_{B_i} : B_i \subset B \text{ and } |B_i| = 2^{-m}|B|\}$ and $q_i = U_{B_i}$ with indices $i \in \mathbb{N}$. It is easy to see that for every $q_i$, we have $d_{\text{TV}}(p, q_i) \geq p(\text{supp}(p) \setminus \text{supp}(q_i)) = \frac{|B| - 2^{-m}|B|}{|B|} = 1 - 2^{-m}$. However, if we consider the distribution $Q = U_{\mathcal{C}'}$, the distribution $|Q|^m$ generates a sample by first producing a distribution $q_i$ which is uniform over some random subset set $B_i \subset B$ with $|B_i| = 2^{-m}|B|$ and then sampling $S \sim q_i^m$. Note that since $B_i$ was selected uniformly at random and $q_i = U_{B_i}$, every point $x \in B$ has the same probability of appearing in a sample $S \sim |Q|^m$. Similarly, every point $x \in B$ has the same probability of appearing in a sample $S' \sim p^m$. Thus, $p^m$ and $|Q|^m$ cannot be distinguished from samples with no repeating elements. While samples from $|Q|^m$ are much more likely to contain repeated elements (as the subset $B_i$ from which they are selected is much smaller than the set $B$), the likelihood of repeated elements appearing in $S \sim |Q|^m$ is still very small. In particular, the probability of there being repeated instances in $S \sim q_i^m$ is upper bounded by $1 - \left(1 - \frac{1}{2^{-m}|B|}\right) \cdot \ldots \cdot \left(1 - \frac{m-1}{2^{-m}|B|}\right) < 1 - \left(1 - \frac{m}{2^{-m}|B|}\right)^m = 1 - \left(1 - \frac{m}{2^{-m}\left(\frac{2^m m}{1-(1-\zeta)^{1/m}}\right)}\right)^m =$

$1 - (1-\zeta)^{1/m})^m = \zeta$ by the birthday problem. Thus, the probability of distinguishing $|Q|^m$ from $p^m$ can be arbitrarily small, i.e., $d_{\text{TV}}(p^m, |Q|^m) < \zeta$ despite the large TV-distance between $p$ and every $q_i$. This suffices to show that any learner $A$ there exists $q \in \mathcal{C} \cup \{p\}$ such that $A$ will not succeed to output a distribution with $d_{\text{TV}}(A(S), q) < \frac{1}{2} - 2^{-m-1}$ on more than $\frac{1}{2} - \frac{\zeta}{2}$ of the proportion of samples $S \sim q^m$.

# C. Alternative proof for $f$-robust learning

We now consider a more general version of robust learning, namely a version that allows the impact of the budget $\eta$ to impact the guarantee via a general function $f$, rather than just being scaled linearly for some $\alpha \geq 1$.

In particular, we are considering $f$ meeting the following requirements.

- $f(0)=0$

- $f$ is continuous

- $f$ is monotonously increasing.

The guarantee for $f$-robust learning is then a generalization of $\alpha$-robust learning, where we can think of $\alpha$-robust learning as the version where $f$ is a linear function. That is $f$-robust learning considers the following learning guarantee:

$$d_{\mathrm{TV}}(A(V(S),p) \leq f(\eta) + \varepsilon.$$

Hence, we get the following definition.

**Definition C.1** (adaptive $f$-robust with respect to adversary $V$)**.** Let $f : [0,1] \to [0,1]$. A class of distributions $\mathcal{C}$ is adaptively $f$-robustly learnable w.r.t. adversary $V$, if there exists a learner $A$ and a sample complexity function $m_{\mathcal{C}}^{V,f} : (0,1)^2 \to \mathbb{N}$, such that for every $p \in \mathcal{C}$, every $\varepsilon, \delta \in (0,1)$ and every sample size $m \geq m_{\mathcal{C}}^{V,f}(\varepsilon, \delta)$ with probability $1 - \delta$,

$$d_{\mathrm{TV}}(A(V(S)),p) \leq f(\mathrm{budget}(V)) + \varepsilon.$$

**Theorem C.2.** *Let $\mathcal{C}$ be a class of distributions and $\mathcal{V} \supset \{V_1, V_2\}$ a set of adaptive adversaries with budgets $\mathrm{budget}(V_1) = \eta_1$ and $\mathrm{budget}(V_2) = \eta_2$. Let $\gamma', \zeta \in (0,1)$ and define*

$$\gamma_f = 2\max\{f(\eta_1), f(\eta_2)\} + 2\gamma'.$$

*If for every $m \in \mathbb{N}$ the pair of adversaries $(V_1, V_2)$ successfully $(\gamma_f, \zeta)$-confuses $\mathcal{C}$-generated samples of size $m$, then $\mathcal{C}$ is not $f$-robustly learnable with respect to $\mathcal{V}$.*

*Furthermore, if $V_1 = V_2$, then $V_1$ is a universal $f$-adversary for $\mathcal{C}$.*

We will now state the alternative proof for this version.

*Proof.* Assume by way of contradiction that there was a successful $\alpha$-robust learner $A$ with sample complexity $m_{\mathcal{C}}$ for $\mathcal{C}$ with respect to $\mathcal{V} \supset \{V_1, V_2\}$.

Let

$$\delta = \frac{1 - \zeta}{2}$$

and

$$\varepsilon = \gamma'.$$

Furthermore, let $m = m_{\mathcal{C}}^{\mathcal{V},\alpha}(\varepsilon, \delta)$. According to the assumptions of the theorem, we know that the pair $(V_1, V_2)$ successfully $(\gamma^f, \zeta)$-confuses $\mathcal{C}$-generated samples of size $m$ with

$$\gamma_f = 2 \cdot \max\{f(\eta_1), f(\eta_2)\} + 2\gamma'.$$

Now consider

$$f(\eta_1) + \varepsilon = f(\eta_1) + \gamma'$$
$$\leq \frac{\gamma_f}{2}$$

With the same argument, we have $f(\eta_1) + \varepsilon \leq \frac{\gamma_f}{2}$. Now using Lemma 5.3, we can infer that there is a distribution $r \in \mathcal{C}$ such that either

$$\mathbb{P}_{S \sim r^m}\left[d_{\mathrm{TV}}(A(V_1(S)), r) > f(\eta_1) + \varepsilon\right]$$
$$\geq \mathbb{P}_{S \sim r^m}\left[d_{\mathrm{TV}}(A(V_1(S)), r) > \frac{\gamma_f}{2}\right]$$
$$\geq \frac{1}{2} - \frac{\zeta}{2} = \delta.$$

or

$$\mathbb{P}_{S \sim r^m}\left[d_{\mathrm{TV}}(A(V_2(S)), r) > \alpha \cdot f(\eta_2) + \varepsilon\right]$$
$$\geq \mathbb{P}_{S \sim r^m}\left[d_{\mathrm{TV}}(A(V_2(S)), r) > \frac{\gamma_f}{2}\right]$$
$$\geq \frac{1}{2} - \frac{\zeta}{2} = \delta.$$

This is a contradiction to the assumption that $A$ is a $f$-robust learner of $\mathcal{C}$ w.r.t $\mathcal{V}$ with sample complexity $m_{\mathcal{C}}^{\mathcal{V},f}$. Furthermore, if $V_1 = V_2$, then $V_1$ is a universal $\alpha$-adversary.

$\square$

**Theorem C.3.** *For every continuous, strictly monotoneously increasing function $f : [0, 1] \to [0, 1]$ with $f(0) = 0$. There is a class $\mathcal{C}$ such that$\mathcal{C}$ is realizably learnable, but not adaptively additive $f$-robustly learnable, nor adaptively subtractive $f$-robustly learnable.*

We now give a more formal version of this statement using the class $\mathcal{C}_g$ from previous section and describing a relation between $f$ and $g$ that is sufficient for $\mathcal{C}_g$ to be a realizably learnable class that is not $f$-robustlly learnable (for both the adaptive additive and adaptive subtractive case).

**Theorem C.4.** *Let $f : [0, 1] \to [0, 1]$. For every function $g : \mathbb{R} \to \mathbb{R}$, for every monotone function $g : \mathbb{N} \to \mathbb{N}$ with $\lim_{n \to \mathbb{N}}$, for which there exists $j \in \mathbb{N}$ such that*

$$\frac{1}{2j} \geq 4f\left(\frac{32}{g(j)}\right) + \frac{2}{g(j)}$$

*for there exists a class $\mathcal{C}_g$ with*

- *$\mathcal{C}_g$ is realizably learnable with sample complexity $m_{\mathcal{C}_g}^{\mathrm{re}}(\varepsilon, \delta) \leq \log\left(\frac{1}{\delta}\right) g\left(\frac{1}{\varepsilon}\right)$*

- *$\mathcal{C}_g$ is not adaptively additive $f$-robustly learnable. Moreover, there is an adaptive additive adversary $V_{\mathrm{add}}$, that is a universal $f$-adversary for $\mathcal{C}_g$.*

- *For every $\alpha \geq 1$, $\mathcal{C}_g$ is not adaptively subtractively $f$-robustly learnable. Moreover, there is an adaptive subtractive adversary $V_{\mathrm{sub}}$, that is a universal $f$-adversary for $\mathcal{C}_g$.*

*Proof.* We note, that it is sufficient to show that there exists $\eta \in [0, 1]$ such that there is an adversary $V_{\mathrm{sub}}$ with budget $\eta$, such that there are $\gamma', \zeta \in (0, 1)$, such that for every $m \in \mathbb{N}$, the adversary $V_{\mathrm{sub}}$ successfully $(4f(\eta) + 4\gamma', \zeta)$-confuses $\mathcal{C}$-generated samples of size $m$. According to our assumption on $g$, there exists $j$ such that

$$\frac{1}{2j} \geq f\left(\frac{32}{g(j)}\right) + \frac{2}{g(j)}$$

Now choose $\eta := \frac{32}{g(j)}$, $\gamma' := \frac{1}{4g(j)}$ and $\zeta = \frac{1}{8}$.

As in the proof of Theorem 4.1 we consider the class $\mathcal{C}_g$ and the distribution $p = p_{i,j,k} \in C_g$ with $|B_i| = 2^{2^n}$ and a set of distributions $D_{i,n,j,g(j)} = \{p_{i',j,k} : B_{i'} \subset B_i : |B_{i'}| = 2^n\}$. As in the previous proof, we note that for every $q \in D_{i,n,j,k}$

we have

$$
\begin{aligned}
d_{\mathrm{TV}}(p, q) &\geq \left(\frac{1}{j} - \frac{1}{g(j)}\right) d_{\mathrm{TV}}(U_{B_i \times 2j}, U_{B_{i'} \times 2j}) \\
&\geq \frac{1}{2}\left(\frac{1}{j} - \frac{1}{g(j)}\right).
\end{aligned}
$$

Using our assumption on the relation between $f$ and $g$ as well as the definition of $j$, we can further infer that.

$$
\begin{aligned}
d_{\mathrm{TV}}(p, q) &\geq \left(\frac{1}{j} - \frac{1}{g(j)}\right) d_{\mathrm{TV}}(U_{B_i \times 2j}, U_{B_{i'} \times 2j}) \\
&\geq \frac{1}{2}\left(\frac{1}{j} - \frac{1}{g(j)}\right) \\
&\geq 4f\left(\frac{32}{g(j)}\right) + \frac{1}{g(j)} \\
&= 4f(\eta) + 4\gamma'.
\end{aligned}
$$

Now if we pick the meta-distribution $Q$ as the uniform distribution over the set $D_{i,n,j,g(j)}$ with $n := \frac{m}{1 - \left(1 - \frac{1}{32}\right)^{\frac{1}{m}}} + 1$. By the same calculation as in the proof of Theorem 4.1, we get

$$
d_{\mathrm{TV}}(V_{\mathrm{sub},\eta}(p^m), V_{\mathrm{sub},\eta}(|Q|^m)) \leq \frac{1}{8}.
$$

Thus, for every $m \in \mathbb{N}$ the adversary $V_{\mathrm{sub}}$ successfully $(4f(\eta) + 4\gamma', \zeta)$-confuses $\mathcal{C}_g$ generated samples of size $m$. Using Theorem 7.1 and Theorem C.2, we get the claimed result.

$\square$

Since for every monotonously increasing, continuous $f : [0, 1] \to [0, 1]$ with $f(0) = 0$ and every $j \in \mathbb{N}$, there exists some $\eta \in [0, 1]$, such that $f(\eta) < \frac{1}{4j}$, it is furthermore possible to define a function $g : \mathbb{N} \to \mathbb{N}$ such that $g(j) \geq \max\{\frac{32}{\eta}, 4j\}$. It follows that,

$$
\begin{aligned}
\frac{1}{2j} &\geq \frac{1}{4j} + \frac{1}{4j} \\
&\geq f(\eta) + \frac{1}{g(j)} \\
&\geq f\left(\frac{32}{g(j)}\right) + \frac{1}{g(j)}.
\end{aligned}
$$

Thus, for every monotonoulsy increasing, continuous function $f : [0, 1] \to [0, 1]$ with $f(0) = 0$, there exists a class $\mathcal{C}$, such that $\mathcal{C}$ is PAC learnable in the realizable case, but not adaptively additively $f$-robustly learnable, nor adaptively subtractively $f$-robustly learnable.

However, it might not be the case that there is a universal counterexample that holds true for all functions continuous, monotonously increasing functions $f$ with $f(0) = 0$ simultaneously. Whether this is the case remains an open question (for each of the following versions of robustness: adaptive additive, adaptive subtractive and oblivious subtractive).

## D. Oblivious Hardness implies Adaptive Hardness

In the introduction, we argued that oblivious subtractive hardness immediately implies adaptive subtractive hardness. In this section, we want to make this claims precise. The adaptive adversaries that we consider in this section have access to both a sample $S$ and the ground-truth distribution $p$. Thus, they differ from the adaptive adversaries that we discussed in the main body of the paper which only required access to $S$.

We want to argue that every successful oblivious subtractive adversary also defines a successful adaptive subtractive adversary. Intuitively, any oblivious adversary has only access to the ground-truth distribution $p$ can be viewed as an adaptive adversary that does not use any knowledge of the sample $S$. However, in order to make this claim precise, we first need to formally define oblivious adversaries. We also need to address the fact that the outputs of oblivious and adaptive adversaries are of different types and thus not equivalent: Oblivious adversaries take as input a ground truth distribution $p$ and output a manipulated distribution $p'$, while subtractive adaptive adversaries take as input a sample $S$ and output a subset of $S' \subset S$.

**Oblivious Adversary**  An *oblivious adversary* $V_{\mathrm{obl}} : \Delta(\mathcal{X}) \to \Delta(\mathcal{X})$ is a function that maps a ground-truth distribution $p$ to some manipulated distribution. When learning in the presence of oblivious adversary $V_{\mathrm{obl}}$ the training sample is i.i.d. sampled from the manipulated distribution $V_{\mathrm{obl}}(p)$.

**Budget**  The budget of an oblivious adversary $V_{\mathrm{obl}}$ is defined by $\mathrm{budget}(V_{\mathrm{obl}}) = \sup_{p \in \Delta(\mathcal{X})} d_{\mathrm{TV}}(p, V_{\mathrm{obl}}(p))$.

**Additive Oblivious Adversaries**  An oblivious adversary $V_{\mathrm{obl}}^{\mathrm{add}}$ is additive with fixed budget $\eta$, if for every $p \in \Delta(\mathcal{X})$, there exists some distribution $r \in \Delta(\mathcal{X})$ such that $V_{\mathrm{obl}}^{\mathrm{add}}(p) = (1 - \eta)p + \eta r$. It is easy to see that using the budget definition from above, we indeed have $\mathrm{budget}(V_{\mathrm{obl}}^{\mathrm{add}}) = \sup_{p \in \Delta(\mathcal{X})} d_{\mathrm{TV}}(p, V_{\mathrm{obl}}^{\mathrm{add}}(p)) \leq \eta$.

**Subtractive Oblivious Adversaries**  An oblivious adversary $V_{\mathrm{obl}}^{\mathrm{sub}}$ is subtractive with fixed budget $\eta$, if for every $p \in \Delta(\mathcal{X})$, there exists some distribution $r \in \Delta(\mathcal{X})$ such that $p = (1 - \eta)V_{\mathrm{obl}}^{\mathrm{sub}}(p) + \eta r$. Similar to above, we have $\mathrm{budget}^{(}V_{\mathrm{obl}}^{\mathrm{sub}}) = \sup_{p \in \Delta(\mathcal{X})} d_{\mathrm{TV}}(p, V_{\mathrm{obl}}^{\mathrm{sub}}(p)) \leq \eta$.

**Learnability with respect to Oblivious Adversaries**  A class $\mathcal{C}$ of distributions is $\alpha$-robustly learnable with respect to a class of oblivious adversaries $\mathcal{V}$ if there is a learner $A : \mathcal{X}^* \to \Delta(\mathcal{X})$ and function $m_{\mathcal{C}}^{\mathcal{V}} : (0,1)^2 \to \mathbb{N}$, such that for every $\varepsilon, \delta \in (0,1)$, for every $p \in \mathcal{C}$ and every $V \in \mathcal{V}$, for every $m \geq m_{\mathcal{C}}^{\mathcal{V}}(\varepsilon, \delta)$ with probability $1 - \delta$ over $S \sim V(p)^m$ we have
$$d_{\mathrm{TV}}(A(S), p) \leq \alpha \cdot \mathrm{budget}(V) + \varepsilon.$$

If a class $\mathcal{C}$ is $\alpha$-robustly learnable with respect to the class of all oblivious adversaries, it is said to be $\alpha$-*robustly learnable*. If a class $\mathcal{C}$ is $\alpha$-robustly learnable with respect to the class of all additive oblivious adversaries, it is said to be *additive $\alpha$-robustly learnable*. If a class $\mathcal{C}$ is $\alpha$-robustly learnable with respect to the class of all subtractive oblivious adversaries, it is said to be *subtractive $\alpha$-robustly learnable*.

We now argue, that given a successful subtractive oblivious adversary $V_{\mathrm{obl}}$, it is possible to define a successful (ground-truth aware) subtractive adaptive adversary $V_{\mathrm{adp}} : \mathcal{X}^* \times \Delta(\mathcal{X}) \to \mathcal{X}^*$.

**Theorem D.1.** *Given a subtractive oblivious adversary $V_{\mathrm{obl}}$ with budget $\mathrm{budget}(V_{\mathrm{obl}}) = \eta$, constants $\varepsilon, \delta \in (0,1)$, a sample size $m \in \mathbb{N}$ and a distribution $p \in \Delta(\mathcal{X})$ such that*

$$\mathbb{P}_{S \sim V_{\mathrm{obl}}(p)^{m - \lceil \eta \cdot m \rceil}}[d_{\mathrm{TV}}(A(S), p) > \alpha \cdot \mathrm{budget}(V_{\mathrm{obl}}) + \varepsilon] > \delta.$$

*Then there is a subtractive adaptive adversary $V_{\mathrm{adp}} : \mathcal{X}^* \times \Delta(\mathcal{X}) \to \mathcal{X}^*$, with (adaptive) budget $\frac{\lceil m\eta \rceil}{m} \approx \eta$, such that*

$$\mathbb{P}_{S \sim p^m}[d_{\mathrm{TV}}(A(V_{\mathrm{adp}}(S)), p) \leq \alpha \cdot \mathrm{budget}(V_{\mathrm{adp}}) + \varepsilon] > \frac{\delta}{2}.$$

*Proof.* Since $V_{\mathrm{obl}}$ is subtractive with budget $\eta$, there is $r \in \Delta(\mathcal{X})$ with $p = (1 - \eta)V_{\mathrm{obl}}(p) + \eta r$. We want to define a subtractive adaptive adversary $V_{\mathrm{adp}}$ that takes into account $p$ and $r$ in such a way that it fulfills the requirement. We thus need to specify a way in which elements of a randomly drawn sample are deleted. Let $\perp$ denote an abstract element that is not element of $\mathcal{X}$. An instance of $\perp$ can be thought of as a "deleted element". We now define an element-wise randomized subproceedure $\mathrm{ElementRandomDelete} : \mathcal{X} \times \Delta(\mathcal{X}) \times \Delta(\mathcal{X}) \to \mathcal{X} \cup \{\perp\}$ that randomly deletes $x$ according to its probability (or density in the continuous case) of $p(x)$ and $r(x)$ respectively:

$$\mathrm{ElementRandomDelete}(x, p, r) = \begin{cases} x & \text{, with probability } \frac{p(x) - \eta \cdot r(x)}{p(x)} \\ \perp & \text{, with probability } \frac{\eta \cdot r(x)}{p(x)} \end{cases}$$

Now for a sample $S = \{x_1, \ldots, x_m\}$, we define the corresponding operation $\mathrm{SampleRandomDelete} : \mathcal{X}^* \times \Delta(\mathcal{X}) \times \Delta(\mathcal{X}) \to (\mathcal{X} \cup \{\bot\})^*$ as element-wise (and independent) application of $\mathrm{ElementRandomDelete}$:

$$\mathrm{SampleRandomDelete}(S, r, p) = \mathrm{SampleRandomDelete}(\{x_1, \ldots, x_m\}, p, r) :=$$
$$= \{\mathrm{ElementRandomDelete}(x_1, p, r), \ldots, \mathrm{ElementRandomDelete}(x_m, p, r)\}.$$

Now consider a sample $S \sim p^m$. We now want to understand the distribution of $\mathrm{SampleRandomDelete}(S, p, r)$. First, we note that since $\mathrm{SampleRandomDelete}$ applies $\mathrm{ElementRandomDelete}$ on all elements independently, we have $\mathrm{SampleRandomDelete}(S, p, r) = \bigcup_{x \in S} \mathrm{ElementRandomDelete}(x, p, r)$, where every $x \in S$ is independently drawn according to $p$. Now, the distribution $q$ of $\mathrm{ElementRandomDelete}(x, p, r)$ for $x \sim p$ can be understood as follows for $x' \in \mathcal{X}$, we have

$$q(x') = p(x') \cdot \frac{p(x') - \eta \cdot r(x')}{p(x')} = p(x') - \eta \cdot r(x') = (1 - \eta)V_{\mathrm{obl}}(p)(x'),$$

since $\mathrm{ElementRandomDelete}(x, p, r)$ can only elvaluate to $x'$ if $x' = x$. Furthermore, for $q(\bot)$ we get

$$q(\bot) = \int_{x \in \mathcal{X}} p(x) \frac{\eta \cdot r(x)}{p(x)} = \int_{x \in \mathcal{X}} \eta \cdot r(x) = \eta.$$

Thus, $\mathrm{SampleRandomDelete}(S, p, r)$ is distributed according to $q^m$, where $q = (1 - \eta)V_{\mathrm{obl}}(p) + \eta \delta_{\{\bot\}}$, where $\delta_{\{\bot\}}$ is the deterministic distribution with all its mass on $\{\bot\}$. The distribution $q^m$ can alternatively be understood as $V_{\mathrm{obl}}(p)^{m-n} \times \delta_{\{\bot\}}^{(}n)$ for a binomial random variable $n \sim \mathrm{Binom}(m, \eta)$. Since the median of a binomial distribution $\mathrm{Binom}(m, \eta)$ is between $\lfloor \eta m \rfloor$ and $\lceil \eta m \rceil$, with probability greater $\frac{1}{2}$ we have $n \leq \lceil \eta m \rceil$. We now define the (randomized) subtractive adaptive adversary $V_{\mathrm{adp}}$ as follows.

- The adversary that first applies $\mathrm{SampleRandomDelete}(\cdot, p, r)$ to $S$ to generate some sample $S'$.

- It then checks, whether $|S' \cap \mathcal{X}| \leq (1 - \frac{\lceil \eta m \rceil}{m})|S|$.

- Case 1: $|S' \cap \mathcal{X}| \leq (1 - \frac{\lceil \eta m \rceil}{m})|S|$. Then in order to match the desired budget, the adversary selects a subset $S'' \subset S \setminus S'$ uniformly at random, such that $|S''| + |S' \cap \mathcal{X}| = ((1 - \frac{\lceil \eta m \rceil}{m}))|S|$ and outputs $S'' \cup (S' \cap \mathcal{X})$.

- Case 2: $|S' \cap \mathcal{X}| \geq (1 - \frac{\lceil \eta m \rceil}{m})|S|$. In this case, the adversary outputs a subset $S''' \subset (S' \cap \mathcal{X})$ which is uniformly selected at random and has size $|S'''| = ((1 - \frac{\lceil \eta m \rceil}{m}))|S|$.

By definition, the adaptive adversary $V_{\mathrm{adp}}$ has a budget of $\mathrm{budget}^{\mathrm{sub}}(V_{\mathrm{adp}}, m) = \frac{m - ((1 - \frac{\lceil \eta m \rceil}{m}))|S|}{m} = \frac{\lceil \eta m \rceil}{m} \approx \eta$.

Lastly, we need to argue that

$$\mathbb{P}_{S \sim p^m}[d_{\mathrm{TV}}(A(V_{\mathrm{adp}}(S)), p) \leq \alpha \cdot \mathrm{budget}(V_{\mathrm{adp}}) + \varepsilon] > \frac{\delta}{2}.$$

We note, that the number of initially deleted elements $|S| - |S' \cap \mathcal{X}|$ corresponds to the previously introduced binomial random variable $n$. As argued before $|S| - |S' \cap \mathcal{X}| = n \leq \lceil \eta m \rceil$ with probability at least $\frac{1}{2}$. Thus with probability at least $\frac{1}{2}$ Case 2 occurs, i.e., $|S' \cap \mathcal{X}| \geq (1 - \frac{\lceil \eta m \rceil}{m})|S|$. Furthermore, conditioned on Case 2 occuring, $V_{\mathrm{adp}}(S)$ is distributed according to $V_{\mathrm{obl}}(p)^{m - \lceil \eta \cdot m \rceil}$. The assumption that

$$\mathbb{P}_{S \sim V_{\mathrm{obl}}(p)^{m - \lceil \eta \cdot m \rceil}}[d_{\mathrm{TV}}(A(S), p) > \alpha \cdot \mathrm{budget}(V_{\mathrm{obl}}) + \varepsilon] > \delta,$$

therefore implies

$$\mathbb{P}_{S \sim p^m}[d_{\mathrm{TV}}(A(V_{\mathrm{adp}}(S)), p) \leq \alpha \cdot \mathrm{budget}(V_{\mathrm{adp}}) + \varepsilon] > \frac{\delta}{2}.$$

$\square$

**Corollary D.2.** *If a class of distributions $\mathcal{C}$ is not subtractive $\alpha$-robustly learnable (in the oblivious case), it is also not adaptively subtractive $\alpha$-robustly learnable.*

This result directly follow from the previous result.

