# OpenReview forum: "On the Learnability of Distribution Classes with Adaptive Adversaries"
_ICML.cc/2025/Conference — ICML 2025 poster_

### Official Review · Reviewer_Xzyp · 2025-03-06

**Overall Recommendation:** 5

**Summary:**

In *robust distribution learning*, the goal is to learn a distribution p in some known class C given samples from p. The goal is to learn p up to some small total variation distance -- that is, find some hypothesis distribution q such that TV(p,q) is small. The twist is that a small fraction of the samples you receive have been modified by some malicious adversary. Robust learning and robust high-dimensional statistics have received extensive attention in machine learning and theoretical computer science in the last decade.

A basic question is: how does the power of the adversary affect which distribution classes are learnable? A key distinction is between *oblivious* and *adaptive* adversaries. An oblivious adversary only gets to modify the distribution p itself, but the samples are still drawn iid from the modified distribution. An adaptive adversary sees the whole list of iid samples from p and then selects a small fraction of them to modify. It's clear that an adaptive adversary is at least as powerful as an oblivious one. This paper addresses the fundamental question: are adaptive adversaries strictly more powerful?

They answer this affirmatively, by constructing a class of distributions C and proving that C is learnable in the presence of an oblivious adversary but not in the presence of an adaptive one.

**Claims And Evidence:**

Yes

**Essential References Not Discussed:**

none

**Experimental Designs Or Analyses:**

N/A

**Methods And Evaluation Criteria:**

Yes

**Other Comments Or Suggestions:**

none

**Other Strengths And Weaknesses:**

The paper is very well written and addresses a fundamental question in robust learning.

**Questions For Authors:**

none

**Relation To Broader Scientific Literature:**

This paper has extensive connections to the literature on agnostic learning, distribution learning, and robust statistics. They are addressed capably in the paper and I will not repeat them here. The most salient connection is to a recent line of work directly addressing the difference between adaptive and oblivious adversaries for distribution learning. This paper answers a significant open question from that literature.

**Theoretical Claims:**

Did not check.

---

> ### Author Rebuttal · Authors · 2025-04-01
>
> Thank you for the positive review! We are happy to hear that the reviewer appreciates the high quality of our writing and the significance of the open problem we resolve.

---

### Official Review · Reviewer_grGP · 2025-03-08

**Overall Recommendation:** 3

**Summary:**

Post rebuttal: The authors addressed my main concerns, and so I have raised my score by one point.
--
This paper studies the question of whether learnability, or realizable learnability, in the PAC sense implies learnability against an *adaptive adversary*. There are multiple notions of adaptive adversaries, and the notion studied in this paper is the one popular in robust statistics. The adversary here receives i.i.d. samples from the distribution to be learned, and then can add some $\epsilon$-fraction of arbitrary samples (not constrained to being sampled from some distribution), or remove some $\epsilon$-fraction of the samples. The first type of adversary is called an *additive* adversary whereas the second type is called a *subtractive* adversary.

Previous work of Ben-David, Bie, Kamath, and Lechner (NeurIPS'23) studied a similar question when the adversary is oblivious. An oblivious adversary is one that is able to change the distribution by some bounded amount (in TV distance) *before* samples are generated, without the ability to edit the samples after seeing them. The work of BBKL shows that learnability implies robust learnability (against the oblivious adversary) in the additive case, but not in the subtractive case. For adaptive adversaries, the non-learnability in the subtractive case is implied by BBKL, but they left open (and explicitly mentioned as an open question) the case of adaptive additive adversaries.

The current paper settles this case, showing that -- unlike in oblivious adversaries -- offline learnability *does not* imply learnability against adaptive additive adversaries. Along the way, the authors show that additive and subtractive adversaries are equivalent in the adaptive setting (this is not true in the oblivious setting). The construction showing the non-learnability is an adaptation/modification of the proof of BBKL.

**Claims And Evidence:**

Yes, it is a theoretical paper and the claims are proved.

**Essential References Not Discussed:**

This seems to be one weakness of the paper. The interaction between oblivious and adaptive adversaries in PAC learning has a rich history, for example in the online version of PAC learning it is known that the combinatorial parameter that captures learnability and agnostic learnability is the Littlestone dimension (rather than VC dimension). I encourage the authors to mention more work of this type.

**Experimental Designs Or Analyses:**

NA - theoretical paper

**Methods And Evaluation Criteria:**

NA - theoretical paper

**Other Comments Or Suggestions:**

See the above bullet. I suggest for the authors to add either a section describing the main ideas in an easier to understand high level description, or include a high level description of each proof at the beginning of the proof.

**Other Strengths And Weaknesses:**

The results of this paper, if new and correct, are significant and worth publishing in a top tier ML venue. The quality of writing is not yet good enough however, and has seemingly prevented me from understanding the main ideas despite trying to put some effort reading into the main ideas. The authors dive straight to the proofs of the results, without providing intuition to the reader, and given that the results are somewhat technical this makes them hard to follow.

**Questions For Authors:**

NA

**Relation To Broader Scientific Literature:**

I do not know of any concrete prior results implying the ones in this paper, but am not an expert on this topic.

**Theoretical Claims:**

I did not check the proofs, and in fact the paper was hard for me to follow from a technical perspective. More below.

---

> ### Author Rebuttal · Authors · 2025-04-01
>
> Thank you for the thoughtful review! We are glad to hear the reviewer appreciates the significance of our results (c.f. “if new and correct, are significant and worth publishing in a top tier ML venue”).
>
> We acknowledge fair criticisms made regarding writing. Indeed, the main critique of the paper is related to the presentation and organization of the paper, rather than the results themselves. These comments are certainly well-received, and we are happy to take them into account in future revisions. Here is a brief overview of the main ideas, which we plan to add to the paper (along with proof descriptions).
>
> **The goal**
>
> Show that adaptive additive adversaries are strictly stronger than oblivious additive adversaries.
>
> **(A) Establish a sufficient criterion for an adversary to render a class of distributions unlearnable [Theorem 4.1]**
>
> We state a condition for an adaptive adversary, that when satisfied, means that it will foil any successful learner for the class. Roughly speaking, the condition asks for an adversary that is capable of taking samples drawn from TV-seperated members of the class and rendering their samples indistinguishable. When class members are separated by large TV distance, any successful learner must distinguish them via the drawn sample. Making use of the idea of a meta-distribution over members of the class, we demonstrate the conflict of the above two statements.
>
> **(B) Unlearnability criterion met for subtractive adversary $\implies$ Unlearnability criterion met for additive adversary [Theorem 5.1]**
>
> Formally, if there is an element $p\in C$ and a meta distribution $Q$ over $C$ that can be made indistinguishable by a subtractive adversary $A_{sub}$, i.e. $d_{TV}(A_{sub}(|Q|^m,A_{sub}(p^m)) < c$, then this adversary can be used to construct additive adversaries $A_{add,p}$ and $A_{add,Q}$, such that the resulting additive sample distributions are close, i.e.
> $d_{TV}(A_{add,p}(|Q|^m),A_{add,Q}(p^m)) < c$.
>
> Adversaries act on samples drawn from two different distributions to make them indistinguishable. However to communicate the main idea, let us talk in terms of simple point-sets rather than distributions.
>
>
> Roughly speaking, the subtractive adversary can remove part of the first sample $A$ or part of the second sample $B$ to leave behind a common coreset $C$. We can view the generative process of $A$ to be a sample from $C$ combined with a sample from $A \setminus C$, and the generative process of $B$ to be a sample from $C$ combined with a sample from $B \setminus C$. Hence to confuse the learner, the additive adversary just needs to add the “opposing piece”: mapping $A \to C + A\setminus C + B \setminus C$ and mapping $B \to C + B\setminus C + A \setminus C$. This introduces indistinguishability with only additions. This intuition is made rigorous in the proof of Theorem 5.1.
>
> **(C) Putting it together: giving a learnable class and constructing a subtractive adversary for it that satisfies the unlearnability criterion [Theorem 6.1]**
>
> We use the same class that Ben-david et al. (2023) used to separate realizable and agnostic learning. It is a class where every member distribution has a unique identifying support element: this makes learning easy if the mass on that support element is high enough to guarantee being observed. It is also constructed to be difficult to learn if those those elements are removed. For this class, we construct an adaptive subtractive adversary that satisfies the unlearnability criterion (A). Via (B), we have a successful adaptive additive adversary. Ben-david et al. (2023) show that realizably learnable classes are learnable in the presence of oblivious additive adversaries, thus completing the separation.
>
> > Oblivious and adaptive adversaries in PAC learning has a rich history… I encourage the authors to mention more work of this type.
>
> Thank you for pointing that out. We reference Blanc & Valiant (2024) which studies the PAC setting and includes a detailed review, but mistakenly did not mention these works. We'll update:
>
> In the PAC setting, the study of oblivious and adaptive adversaries has a rich history. The agnostic setting of Hausler (1992) is the PAC analogue to learnability in the presence of *oblivious adversaries* in our setting. Indeed, in the PAC setting, learnability implies oblivious-adversary-robust learnability. The PAC analogue of the *adaptive adversaries* of the present paper has been studied under the name "nasty noise" (Bshouty et al, 2002). The malicious noise model, introduced by Valiant (1985) and studied in more generality by Kearns and Li (1988), can be viewed as a partially adaptive adversary (see Blanc and Valiant (2024) for further elaboration); it has inspired significant follow-up work (Klivans et al. 2019, Aswathi et al. 2017). For a more careful treatment of learning in the presence of adversaries in the PAC setting, we refer the reader to the excellent review of Blanc & Valiant (2024).

---

> > ### Comment · Reviewer_grGP · 2025-04-02
> >
> > Thank you for the response. I will raise my score by one point.

---

### Official Review · Reviewer_TGTy · 2025-03-14

**Overall Recommendation:** 4

**Summary:**

The paper investigates the problem of learning distributions under an adaptive adversary, i.e., adversary has full knowledge of sample and can apply noise to the samples before passing it to the learner.

**Claims And Evidence:**

Main result (MR): There is a class of distributions C that is not learnable in the realizable setting in the presence of adaptive adversaries.

C1. In my understanding Theorem 4.1, formally captures the fact that if adverseries are powerful enough to make seperated distributions (w.r.t TV, first two conditions of the theorem), close to each other (third condition of the theorem) then learning is not possible.

C2.  Theorem 5.1 proves that adaptive subtractive adversaries can be “translated” into the additive setting, thereby establishing that if robust learning fails against subtractive adversaries, it also fails against additive adversaries.

C3. Theorem 6.1 shows that there exists a class of distributions that is learnable but becomes unlearnable under adaptive adversaries.

The paper provides quite intuitive proofs and on of the main strategies is explained in section 4.

**Essential References Not Discussed:**

None

**Experimental Designs Or Analyses:**

NA

**Methods And Evaluation Criteria:**

The authors use a general (quite intuitive) existing strategy (Lemma 4.2). Given two distribution classes C_1 and C_2, such that all distribution (p and q respectively) in the two classes are far apart in TV. And two adversaries, V_1 and V_2 acting on a distribution over C_1  and p \in C_2, then for any learner there is atleast one distribution in C_1 \cup C_2 such that learning it is not possible.

**Other Comments Or Suggestions:**

See above

**Other Strengths And Weaknesses:**

Clarity: I think the authors have provided a very clear presentation in terms of ordering and framing of the theoretical results. However, I feel some toy examples could further aid clarity.

Novelty: I found the ideas in the paper quite interesting and the problem is also quite relevant and natural.

**Questions For Authors:**

- How critical is the superlinear growth of g? Are there classes with milder growth for which a similar separation might hold?

- Do you have any intuition about what minimal properties of a distribution class can make it more robust to adversaries as defined in your setting?

- Do you think these results fundamentally rely on TV, being somewhat of a strict distance? maybe achieving adversaries for Wasserstein Distance is harder?  --- just a curiosity.

**Relation To Broader Scientific Literature:**

The paper addresses an important problem of distribution learning. And clearly presents the broader connection to general work in distribution learning.

**Theoretical Claims:**

See Claims and Evidences.
I did not check all the proofs.
But the theorems are clearly written and intution is relatively easy to fallow. Although, more focus on natural language explanations or a running toy example can help a lot more.

---

> ### Author Rebuttal · Authors · 2025-04-01
>
> Thank you for the positive review! We are glad that the reviewer found the ideas in the paper quite interesting and the problem studied to be quite relevant.
>
> >...the theorems are clearly written and intution is relatively easy to fallow. Although, more focus on natural language explanations or a running toy example can help a lot more…
>
> > I think the authors have provided a very clear presentation in terms of ordering and framing of the theoretical results. However, I feel some toy examples could further aid clarity.
>
> We are happy to hear that overall, the reviewer finds our presentation of results to be clear, intuitive, and easy to follow. Indeed, we acknowledge the benefit of toy examples and more natural language explanations for improved accessibility –  we will incorporate them in the next revision. Please see the reply to the below reviewer for a natural language explanation of the proof ideas in the current discussion period.
>
> > How critical is the superlinear growth of g?
>
> The fact that we need superlinearity comes from the fact that we need the relation between the manipulation budget and the effect on accuracy to scale larger than linear in order to meet the definition for “not learnable”.
>
> > Do you have any intuition about what minimal properties of a distribution class can make it more robust to adversaries as defined in your setting?
>
> Classes that are agnostically learnable should be more likely to be robust to adversaries. As such classes where the Yatracos sets have finite VC-dimension should be robust.
>
> > Do you think these results fundamentally rely on TV?
>
> Our paper only discusses PAC learning with respect to TV-distance, which is a well-established setting in the learning theory community. While some of these effects might transfer to different divergence measures between distributions, our current examples might not go through for measures like KL-divergence, as KL-divergence requires distributions to have the same support to not explode, which is not the case or TV-distance (and is a property that we in our construction exploit).

---

### Decision · Program_Chairs · 2025-05-01

**Decision:**

Accept (poster)

**Comment:**

The paper studies distribution learning with adaptive adversaries in contrast to learning with oblivious adversaries. An adaptive adversary intercept samples and corrupt them before passing them to the learner, while an oblivious adversary can only modify the distribution the samples. Various theoretical results are presented, answering an important open question regarding equivalence of the adaptive and oblivious regimes.

The reviewers unanimously agree to accept this paper. For a camera-ready version, please take into the account the comments from all reviewers regarding for instance improving the writing and adding more references. In spite of this, the reviewers still support acceptance of this paper.